



# When physics gets in the way: an entropy-based evaluation of conceptual constraints in hybrid hydrological models

Manuel Álvarez Chaves[1], Eduardo Acuña Espinoza[2], Uwe Ehret[2], and Anneli Guthke[1]

[1]Stuttgart Center for Simulation Science, Cluster of Excellence EXC 2075, University of Stuttgart, 70569 Stuttgart, Germany
[2]Institute of Water and Environment, Karlsruhe Institute of Technology (KIT), 76131 Karlsruhe, Germany

**Correspondence:** Manuel Álvarez Chaves (manuel.alvarez-chaves@simtech.uni-stuttgart.de)

**Abstract.** Merging physics-based with data-driven approaches in hybrid hydrological modeling offers new opportunities to enhance predictive accuracy while addressing challenges of model interpretability and fidelity. Traditional hydrological models, developed using physical principles, are easily interpretable but often limited by their rigidity and assumptions. In contrast, machine learning methods, such as Long Short-Term Memory (LSTM) networks, offer exceptional predictive performance

but are often criticized for their black-box nature. Hybrid models aim to reconcile these approaches by imposing physics to constrain and understand what the ML part of the model does. This study introduces a quantitative metric based on Information Theory to evaluate the relative contributions of physics-based and data-driven components in hybrid models. Through synthetic examples and a large-sample case study, we examine the role of physics-based conceptual constraints: can we actually call the hybrid model "physics-constrained", or does the data-driven component overwrite these constraints for the sake

of performance? We test this on the arguably most constrained form of hybrid models, i.e., we prescribe structures of typical conceptual hydrological models and allow an LSTM to modify only its parameters over time, as learned during training against observed discharge data. Our findings indicate that performance predominantly relies on the data-driven component, with the physics-constraint often adding minimal value or even making the prediction problem harder. This observation challenges the assumption that integrating physics should enhance model performance by informing the LSTM. Even more alarming, the data-

driven component is able to avoid (parts of) the conceptual constraint by driving certain parameters to insensitive constants or value sequences that effectively cancel out certain storage behavior. Our proposed approach helps to analyse such conditions in-depth, which provides valuable insights into model functioning, case study specifics, and the power or problems of prior knowledge prescribed in the form of conceptual constraints. Notably, our results also show that hybrid modeling may offer hints towards parsimonious model representations that capture dominant physical processes, but avoid illegitimate constraints.

Overall, our framework can (1) uncover the true role of constraints in presumably "physics-constrained" machine learning, and (2) guide the development of more accurate representations of hydrological systems through careful evaluation of the utility of expert knowledge to tackle the prediction problem at hand.



## 1 Introduction

Hydrological models are essential tools for the management of water resources as well as scientific research. Due to their wide range of applications, the motivations and reasons behind the choices that lead to the specific usage of a model over another are not clear and the issue of adequacy in the choice of a model is not typically addressed (Horton et al., 2022). Worryingly, the choice of a model is often relegated to past experience and not adequacy (Addor and Melsen, 2019).

Some authors have argued for the creation of a Community Hydrology Model which could be able to represent different processes at different scales, making it suitable for a wide range of applications, but there are open challenges that need to be addressed before such a model can be developed (Weiler and Beven, 2015). In contrast, other authors support the concept of flexible modeling frameworks that enable users to combine different representations of processes and model constructs (Fenicia et al., 2011; Clark et al., 2008). Using this approach, a unique model can be developed for a specific application, and the issue of model adequacy is addressed by testing multiple models as different working hypotheses (Clark et al., 2011).

### 1.1 Conceptual rainfall-runoff models

So far, the traditional modeling approach has been that of simplified physical concepts in which different compartments in the hydrological cycle are represented by interconnected storage units and these models obey physical principles. Thus, understanding of the physical system is translated into the model and vice-versa, making the models easily interpretable.

Typically catchment scale processes of in a rainfall-runoff model are represented by a reservoir element that can be described by ordinary differential equations (ODEs):

$$\frac{dS(t)}{dt} = f\left(S(t), u(t) | \theta\right) \tag{1}$$

$$Q(t) = g\left(S(t), u(t) | \theta\right) \tag{2}$$

Where $S(t)$ represents the conceptual storage of a reservoir element at time $t$, $u(t)$ is time-dependent forcing data, $Q(t)$ is the response of the reservoir element to the forcing and $\theta$ are the model parameters. Furthermore, $f$ and $g$ are functions that describe the evolution of storage and output with time (Fenicia et al., 2011). These types of models are physically-based because the main driving principle of a model is conservation of mass through Eq. 1.

These very simple principles for conceptual rainfall-runoff models have been adapted into modular modeling frameworks such as FUSE (Clark et al., 2008), Superflex (Fenicia et al., 2011; Dal Molin et al., 2021) and RAVEN (Craig et al., 2020). These frameworks enable researchers to develop an unlimited range of modular structures for rainfall-runoff models. In practice, researchers apply these frameworks in model comparison studies using one of typically two approaches: top-down or bottom-up development. The top-down approach begins with a complex model and reduces its components, while the bottom-up approach starts with a simple model and gradually increases its complexity (Hrachowitz and Clark, 2017).





Other studies evaluate an existing set of standard model structures within these frameworks. Although the choice of components is often arbitrary and informed by prior experience, a paradigm for automatic model structure identification has been proposed which systematically tests and identifies the most adequate model structures for a rainfall-runoff model while acknowledging the challenge of equifinality (Spieler et al., 2020).

## 1.2 LSTMs

Unlike the previous approach, machine learning (ML) and other purely data-driven approaches assume no prior knowledge and learn the required relationships between variables from the provided data alone. In particular, Long Short-Term Memory (LSTM) networks have been shown to provide very accurate predictions of streamflow establishing a number of benchmarks across different data sets (Kratzert et al., 2018, 2019a, b; Lees et al., 2021; Loritz et al., 2024). The performance of these models can be partly attributed to the flexibility of LSTM networks (LSTMs hereafter) which do not have the constraints that physically-based models have.

LSTMs (Hochreiter and Schmidhuber, 1997) are a type of recurrent neural network (RNN) which has been widely adapted in hydrology for rainfall-runoff modeling and/or predicting streamflow (Kratzert et al., 2024). More generally, RNNs and LSTMs have found applications in modeling dynamical systems (Gajamannage et al., 2023). Indeed, the reason they have been successful is that this type of neural network adds both memory (that is, states) and feedback to allow for the current output values to depend on past output values and states (Goodfellow et al., 2016). As mentioned previously, because a catchment can be represented as a set of ODEs which make it a dynamical system (Kirchner, 2009), the usage of LSTMs for rainfall-runoff modeling arises naturally. Ultimately, both approaches: conceptual and data-driven models are complementary, and direct mappings between one another have been identified (Wang and Gupta, 2024).

The issue of lacking mass conservation in LSTMs has been addressed by models which include an additional term that accounts for unobserved sinks, pointing towards deficiencies in data products (Frame et al., 2023) and this issue of closure is often a point of discussion and controversy (Beven, 2020; Nearing et al., 2021). Nevertheless, the main criticism of these models comes from their "black-box" nature, which makes their internal processes difficult to understand. Current methods for interpreting neural networks typically require the use of a secondary model to analyze the primary one (Montavon et al., 2018). For example, while researchers have proposed techniques to correlate LSTM hidden states with real-world variables (Lees et al., 2022), this interpretation process remains complex and requires the implementation of an additional model, known as a probe. Some researchers address this challenge by selecting model architectures that are inherently more interpretable, although these approaches still often require supplementary models for comprehensive explainability (De la Fuente et al., 2024).

## 1.3 Hybrid models

Recently, hybrid modeling approaches (Reichstein et al., 2019) have been proposed as an end-to-end modeling system that combines both data-driven approaches with more traditional physics-based models. In particular, differentiable models (Shen et al., 2023) can be considered a subset of hybrid models which take advantage of deep neural networks and differentiable



programming paradigms (Paszke et al., 2019; Bradbury et al., 2018) to calculate gradients with respect to model variables or parameters, enabling the discovery of unknown relationships. These models have been successfully applied in surface hydrology for streamflow prediction across various cases using different approaches (Tsai et al., 2021; Feng et al., 2022). Importantly, current hybrid model applications primarily take advantage of the ability of their data-driven components to exploit information from large datasets, leaving their effectiveness with smaller datasets as an open question. The data requirements

for different hydrological modeling methods remain an active area of research (Kratzert et al., 2024; Staudinger et al., 2025).

The models exemplarily used in our illustrative case studies share similarities with those developed by Feng et al. (2022) and can be classified as differentiable models. In the context of scientific machine learning, they belong to the framework of Universal Differential Equations (UDEs), which combine differential equations with neural networks to represent the dynamics of the system (Rackauckas et al., 2021). The process of solving UDEs allows researchers to identify unknown functions and

system dynamics from data while (presumably) preserving the underlying mathematical structure of the equations.

## 1.4 Key idea

Recent developments show an increasing integration of physics-based and data-driven approaches in hydrological modeling. This trend is evident in streamflow prediction, where researchers have successfully implemented both neural operator-based methods, such as NeuralODEs (Höge et al., 2022), and traditional statistical approaches (Chlumsky et al., 2023). These hybrid

solutions increasingly blur the distinction between purely physics-based and purely data-driven modeling paradigms.

Although this integration is gaining widespread adoption in hydrology, recent work by Acuña Espinoza et al. (2024) raises important questions that need to be addressed. They demonstrate that incorporating physics-based components or prior knowledge doesn't yield an improvement in model performance. Furthermore, hybrid models can perform well even when the incorporated physical principles oversimplify or misrepresent the underlying system, primarily because their data-driven compo-

nents can compensate for these imposed limitations. This observation raises fundamental questions about the value of incorporating physics-based components into hydrological models. While purely data-driven methods often achieve high performance, we lack systematic ways to evaluate when and how the addition of physical principles genuinely enhances model performance and improves the representation of underlying physical processes. This study addresses this knowledge gap through the following contributions:

1. We introduce a quantitative metric to assess whether a hybrid model's performance is dominated by its data-driven or physics-based components;

2. We demonstrate the characteristics of this metric under synthetic conditions, i.e. we guide the modeler's intuition about what to expect if the prescribed constraint is physically meaningful or not;

3. We suggest a diagnostic evaluation routine to better understand the *effective* hybrid model's structure, not its (presum-

ably) *prescribed* one;





4. We derive insights about the relative contribution of physics-based and data-driven components from applying this metric to a large-sample case study, illustrating how "physics may get in the way" under imperfectly known model settings.

In particular, we measure the entropy of both the LSTM-predicted time-variable parameters and the LSTM hidden states to quantify how much the data-driven component of our hybrid model counteracts the conceptual model's prescribed con-
straints. Our hypothesis is that low entropy indicates the LSTM needs minimal parameter variation, suggesting the conceptual constraints accurately describe the natural system. Conversely, high entropy suggests inappropriate constraints (e.g., oversimplified or enforcing mass balance despite imperfect inputs). High entropy points to an imbalance favoring the data-driven component, and subsequent evaluation of LSTM-learned parameters helps determine if this is actually the case. If so, we hope to still identify physical principles within the hybrid model; otherwise, the term "physics-informed" would be proven incorrect.
Our proposed approach helps analyze such conditions in-depth, which provides valuable insights into model functioning, case study specifics, and the strength or limitations of prior knowledge prescribed in the form of conceptual constraints.

We demonstrate our approach through two case studies. The first uses synthetic data with a known "true" model that accurately represents a system, allowing us to test our hypothesis and develop practical insights of our proposed metric. This example builds initial intuition about evaluating hybrid models by measuring entropy in both the conceptual model parameter
space and LSTM hidden state space, and shows how performance can be attributed to either the data-driven or physics-based components. Our second case study applies these insights to a real-world dataset where no "true" model is known, further demonstrating real world and practical application of our metric.

The remainder of the manuscript is structured as follows. Section 2 details the types of models employed in this study, data for the case study and specific aspects of calculating differential entropy in higher dimensions. Section 3 and Sect. 4 cover the
described case studies. Finally, Sect. 5 summarizes our main findings and discusses avenues for future research.

## 2  Data and methods

In this section, we outline the basic elements of our study, including the dataset employed across both case studies, models used, and the general methodological framework for training and evaluation. While this section provides a high-level overview of our methods, the subsequent case-specific sections will discuss more in-depth details, including hyperparameter configurations,
architectural adaptations, data selection criteria, and other specific considerations unique to each experimental scenario.

### 2.1  CAMELS-GB

CAMELS-GB is a large sample catchment hydrology dataset for Great Britain (Coxon et al., 2020). As with similar large-sample datasets (Addor et al., 2017; Loritz et al., 2024), it collects data for streamflow, catchment attributes, and meteorological time-series data for 671 river basins across England, Scotland and Wales.
As in Acuña Espinoza et al. (2024), we based our experimental setup on the approach of Lees et al. (2021). We provide a brief description here and refer readers to these studies as well as Appendix A in this article for further details.





As forcing data, we used the time-series of catchment average values of precipitation, potential evapotranspiration and temperature in the dataset. In addition, as input for the LSTMs, we used 23 of the static attributes that describe the catchments in the dataset. Of these, 3 were related to topography, 6 to soil, 4 to land cover, 1 to human influence and 8 to climate characteristics. These are detailed in Table A3. As part of the experimental setup, the data was divided into training, validation, and testing sets. The training set spans from October 1, 1980, to December 31, 1997; the validation set from October 1, 1975, to September 30, 1980; and the testing set from January 1, 1998, to December 31, 2008.

## 2.2 Models

### 2.2.1 LSTMs

An LSTM is a type of recurrent neural network that effectively addresses the vanishing gradient problem through specialized memory cells with input, forget, and output gates. This architecture enables LSTMs to capture long-term dependencies in sequential data, making them valuable for time series prediction. Their capacity to learn temporal patterns without explicit physical parameterizations has proven particularly effective for modeling streamflow. For a more in-depth description of the applications of LSTMs in hydrology, we refer to the work of Kratzert et al. (2018).

### 2.2.2 Hybrid models

The hybrid models used in our study follow the paradigm of Shen et al. (2023) and combine an LSTM network with a conceptual physics-based representation of the hydrological system. More specifically, our models resemble the proposed $\delta$HBV model (Feng et al., 2022).

In simple terms, this approach to hybrid modeling can be conceptualized as a hydrological model with dynamic parameters. In rainfall-runoff modeling, the use of dynamic parameters originated with data-based mechanistic modeling (Young and Beven, 1994), which established methods for identifying time-*invariant* parameters in relation to their time-*variant* counterparts. More recent approaches generate time-dependent parameters by introducing stochastic processes that represent deviations from calibrated static parameters (Reichert and Mieleitner, 2009). In these methods, both static parameters and their variable components are jointly calibrated via Bayesian updating using Markov chain Monte Carlo. While theoretically convincing, the practical application of stochastic, time-dependent parameters has been very limited due to identifiability problems and the computational burden of propagating time-dependent parameters in a rigorous Bayesian framework (Reichert et al., 2021). With the recent gain in popularity of differentiable models, the idea of dynamic parameters (albeit in a deterministic setting) has experienced a significant revival in hydrological modeling.

Figure 1 shows the models used our case study and the ranges of their parameters are shown in Table A2. At runtime, the LSTM runs for the entire length of a sequence of inputs and predicts the conceptual model's parameters at every time step. These predictions are made in "sequence-to-sequence" mode. After this initial run, the operation of the model resembles a traditional hydrological model with the distinction being that the model reads a new set of parameters at every time step along with it's inputs, therefore the parameters of the model vary in time. Due to the initial run of the LSTM and the warm-up period





of the hydrological model, all hybrid models in this paper use a sequence length of 730 days (2 years) with only the second half

of the predictions ($y^{sim}$) evaluated in the loss function. Furthermore, instead of evaluating the model at each unique selection of 365 time steps, we limit the number of evaluations to 450 chosen randomly, meaning that the loss function is calculated using $365 \cdot 450 = 164\,250$ values of $y^{sim}$ and $y^{obs}$. For more details of the evaluation process, please refer to Acuña Espinoza et al. (2024).

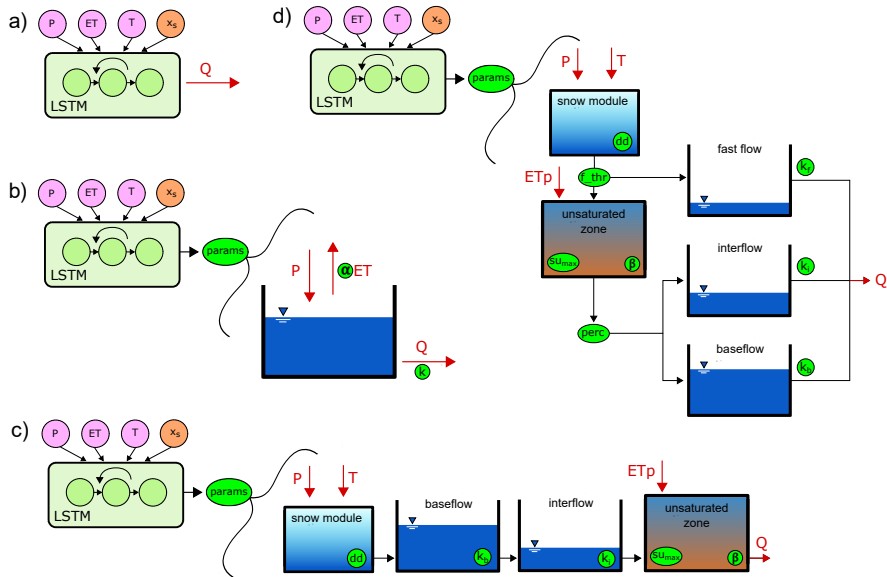

**Figure 1.** Sketch of the hybrid models used in this study. The parameters in each model are encircled and highlighted in green.
a) LSTM, b) Hybrid Bucket, c) Hybrid Nonsense, d) Hybrid SHM.

### 2.2.3 Performance evaluation

Hybrid models are typically trained to make deterministic predictions. Therefore, depending on the case, we use either the mean-squared error (MSE) or the basin average Nash-Sutcliffe efficiency (NSE*) (Kratzert et al., 2019b) as loss functions to evaluate the simulations of our model $y_{sim}$ with the observed data $y_{obs}$.

$$MSE = \frac{1}{N} \cdot \sum_{i=1}^{N} (y_i^{obs} - y_i^{sim})^2 \qquad (3)$$

$$NSE^* = \frac{1}{N} \cdot \sum_{i=1}^{N} \frac{\left(y_i^{obs} - y_i^{sim}\right)^2}{\left(s_i + \epsilon\right)^2} \qquad (4)$$



In Eq. 3, $i$ identifies the predictions and observations on a specific day and $N$ the number of days over which to calculate the loss function. In the case of NSE*, the additional term $s_i$ in Eq. 4 represents the standard deviation of the streamflow time-series and $\epsilon$ is a numerical stabilizer added for cases where $s_i$ is low.

## 2.3    Entropy-based measure of LSTM-induced parameter variability

For our evaluation, we aim to measure how much the LSTM makes the conceptual model's parameters vary over time to achieve
an optimized performance during training. The underlying premise is that, when using a perfect model, constant true parameter values can be found during optimization, and the "LSTM-induced variability" will be zero. If the conceptual constraint is sufficiently honored by the LSTM, we expect a mild or null variability in the predicted timeseries of parameter values. In contrast, if a severely wrong representation of the true system is used as conceptual model, the LSTM will compensate through highly time-dependent parameter values, and the variability in parameters will be high. In Sect. 3.3 we explain the reason why
we don't analyze the variability of the predicted parameter values directly, but of the hidden states of the LSTM.

Although there are several measures of variability, we choose to measure this variability through entropy, as it does not require any assumptions about the type or shape of the statistical distribution of the analyzed data. For analyzing the entropy of timeseries data, we have to evaluate continuous (differential) entropy (Cover and Thomas, 2006; MacKay, 2003) as shown in Eq. 5 with $p$ denoting probability density functions (PDFs) of a random variable $X$ with support $\mathcal{X}$.

$$H(X) = -\int_{\mathcal{X}} p(x) \log p(x) dx, \tag{5}$$

Beirlant et al. (1997) provide a comprehensive overview of different common approaches to estimating differential entropy from data. In this study, we will use the method proposed by Kozachenko and Leonenko (KL) based on nearest neighbor distances (Kozachenko and Leonenko, 1987) shown in Eq. 6:

$$\hat{H}(X) = \psi(N) - \psi(k) + \log(c_1(d)) + \frac{d}{N}\sum_{i=1}^{N}\log(\rho_k^d(i)), \tag{6}$$

where $\psi$ is the digamma function $\frac{d}{dz}log(\Gamma(z))$, $N$ is the number of points in a sample, $k$ us a hyperparameter specifying the number of of nearest neighbors used in the estimate, $c_1(d)$ is the volume of a $d$-dimensional unit ball, $d$ is the number of dimensions of the data and $\rho_k^d(i)$ is the distance between $x_i$ and its $k^{th}$ nearest neighbor. The KL estimator for entropy has been shown to be accurate even for data in higher dimensions (Álvarez Chaves et al., 2024) and was implemented as part of the UNITE toolbox, a suite of tools we have developed for practical applications of information theory in model evaluation.

## 2.4    Diagnostic routine to evaluate hybrid model structure

Analyzing the variability in parameter or hidden-state space highlights cases where the prescribed conceptual constraint fails to accurately reflect the underlying system dynamics. Such discrepancies fall into two categories: cases where the physics





are appropriate but other reasons make the model struggle (e.g., biased or highly uncertain input data), and cases where the physics constraint itself is problematic (e.g., due to neglected or misrepresented processes). To distinguish between these cases

and gain insights into system understanding and model development, we propose a tailored diagnostic evaluation routine that scrutinizes the joint behavior of the LSTM-learned parameters. We demonstrate the effectiveness and diagnostic capabilities of this approach through didactic examples in Sect. 3.

## 3 Didactic examples illustrating the proposed workflow

### 3.1 Motivation

Synthetic examples here serve to create intuition about the role of the data-driven component in hybrid hydrological models. Specifically, we will demonstrate the role of LSTMs predicting time-variant parameters of conceptual hydrological models. We aim to answer the following questions:

1. How does the data-driven component behave in presence of a perfect conceptual constraint (i.e., the physics of the data-generating process are fully reflected in the conceptual model)?

2. How much variability in the LSTM-predicted parameter values will be detectable if the conceptual constraint is reasonable, but not a complete representation of the data-generating process?

3. How will the data-driven component react if the conceptual constraint is not reflecting the data-generating process at all?

Specific details for the experimental setup of these didactic examples are described in Appendix A1. The main point to highlight here is that all models were trained using mean-squared error (MSE) as the loss function (Eq. 3). The reason is that

the data used for $y_{obs}$ was created synthetically by running an initial "true" model; therefore, there was no need to account for differences in the magnitude of the streamflow signals between basins.

As stated in Sect. 1.1, the main principle driving conceptual hydrological models is the conservation of mass in different reservoirs or storages within a model. Following Eq. 1, in a simple case of one storage, conservation of mass can be written as $\frac{dS}{dt} = P - Q - ET$, with $P$ being the input (precipitation) and $ET$ and $Q$ being two outputs (evapotranspiration and out-

put streamflow, respectively). Let us assume a model that represents a typical power reservoir in which the storage-outflow relationship is described by the power function $Q = \frac{S^\beta}{k}$, where $\beta$ and $k$ are model parameters with an additional parameter $\alpha$ being used as a correction factor for the output flux of $ET$. This model and its governing equations are shown in Fig. 2a. Using the precipitation and evapotranspiration time-series of a subset of basins in CAMELS-GB (cf. Sect. 2.1) and parameter values $\alpha = 0.8$, $\beta = 1.2$, $k = 24.0$, we create a synthetic "observed" streamflow time-series that is shown across all plots in Fig. 2

(subplots b, d, f, i, l and o). This initial model is considered our "synthetic truth" because it was used to generate the target data ("observed" streamflow) for the competing formulations of hybrid models described in Sect. 3.2.

We will analyze the resulting time-varying parameter values of the alternative hybrid models, as predicted by their respective LSTM-component, and interpret these results given our knowledge of the true model structure in Sect. 3.3.1. Then, we will





explain how we measure variability as the entropy of the resulting parameter distributions in Sect. 3.3.2, and why we move to
measuring the "activity" of the LSTM in its hidden state space in Sect. 3.3.3. We summarize the key points of our proposed
approach, as illustrated on these didactic examples, in Sect. 3.4.

## 3.2 Hybrid models

To investigate the three research questions posed above, we setup four alternative hybrid models to predict the time-series of
observed discharge, illustrated in the left column of Fig. 2:

1. We use an LSTM to directly predict streamflow from the inputs of precipitation and evapotranspiration (Model 0);

2. We couple the "true" model defined above with an LSTM network to predict its parameters $\alpha$, $\beta$ and $k$, as described in
   Sect. 2.2.2 (Model 1);

3. We substitute the power-reservoir with a linear reservoir that follows the storage-outflow relationship $Q = \frac{S}{k}$, and add
   a threshold parameter $S_{max}$ such that any excess storage directly becomes streamflow $Q = (S - S_{max})$ if $S \geq S_{max}$
   (Model 2);

4. We add an additional reservoir to Model 1 which receives the outflow of the previous reservoir $\frac{dS_1}{dt} = Q_1 - Q_2$ and both
   reservoirs have a linear storage-outflow relationship (Model 3);

5. We extend the storage-outflow relationship of Model 1 with an additional threshold parameter $S_0$ that reflects the mini-
   mum storage required to generate streamflow (Model 4).

Based on the distinction between structures and processes, we have categorized each model according to its architectural
design and process representation. Model 2 maintains the correct one-reservoir architecture of the true model but implements
an incorrect process representation by substituting the true exponential outflow relationship with a simple linear relationship.
Model 3 deviates from the true model in both aspects: it uses the same incorrect linear outflow relationship while also incorpo-
rating an additional storage reservoir that doesn't exist in the true model. Model 4 preserves the correct architecture of the true
model but becomes overparameterized in its process representation by introducing an extra parameter, $S_0$. Interestingly, when
$S_0$ is set to zero, Model 4's process representation perfectly aligns with the outflow relationship in the true model. We explore
these relationships further in Sect. 3.3.1. Additional exemplary model architectures are presented in Appendix B.

For all cases, the coupled LSTM was built and trained in the same way, consisting of ten hidden states, and trained on five
randomly selected basins (76005, 83004, 46008, 50008 and 96001) of the CAMELS-GB dataset. In general, using more than
one basin improved the training process of all models and this was particularly true for the pure LSTM (Model 0), validating
current standard practices (Kratzert et al., 2024). Yet, the purpose of this analysis was not to achieve maximum performance
for a given task, but to compare hybrid approaches on the same grounds. To allow for extensive repetitions and alterations, we
deliberately kept the training effort low (differently from the real-world case study, see Sect. 4).





Note that we only used the *precipitation* and *pet* time series of each basin, and created our own synthetic "observed" stream-
flow as described in Sect. 3.1 for training the models. The train/test split of the data was used as detailed in Sect. 2.1. For
the conceptual components of the models, the ranges allowed for parameter variations are shown in Table A1 (Beck et al.,
2016, 2020). Finally, as the target is the product of a model, there was no need to adjust the loss function to consider spe-
cific characteristics of the data, and hence we chose the MSE (Eq. 3) as loss function. Each model was trained for a specific
number of epochs using specific learning rates for each update between epochs. We suggest the reader to consult the logs of
the synthetic examples to check specifics for each model. The reported metrics for the testing period are averaged over five
realizations of each model obtained from random initializations using different seeds.

### 3.3 Analysis and discussion of results

#### 3.3.1 Visualization of time-varying parameters

For illustration purposes, we show a short five-month period (November 2004 to April 2005, Cumbria and Carlisle floods of
2005 in the UK (Harper, 2015)) in Fig. 2 to demonstrate the ability of all models to perfectly fit the data both during high and
low flow conditions. The center column shows the predicted streamflow by an exemplary run of the hybrid model, and the right
column shows the corresponding parameter trajectories. The reported numerical values of NSE and entropy are for the whole
testing period of January 1, 1998, to December 31, 2008, and averaged over the five runs based on different random seeds for
initialization. The density plots in the right column of Fig. 2 were created using a kernel-based estimate for density (Waskom,
2021), but the reported individual entropies of each parameter and the joint entropy of the model parameters were calculated
using the KL estimator described in Sect. 2.3.

**Perfect physics constraint (Model 1)** In the case of Model 1, where the LSTM is coupled to the true conceptual model, we
hope to see that the data-driven component does nothing, i.e., it doesn't interfere with the perfect representation of the natural
system that is provided by the conceptual constraint. Indeed, we find that the network predicts practically static parameters
as shown in Fig. 2g, with almost negligible deviations only resulting from the effect of the sequential nature of the LSTM.
Reassuringly, the LSTM is able to recover the true parameter values of $\alpha = 0.8$, $\beta = 1.2$, and $k = 24.0$. As a logical conse-
quence, this hybrid model is able to perfectly mimic the observations with an NSE of 1.0, as they were created with the same
conceptual model and parameter values.

**Imperfect physics constraint (Models 2 and 3)** The behavior of the time-varying parameters is expected to differ when
the LSTM is coupled to a conceptual model that does not adequately represent truth. Subplots j) and m) of Fig. 2 illustrate the
behavior of the parameters when the conceptual component of the hybrid model has been incorrectly specified. In these cases,
we can see how the LSTM varies the parameters in order to achieve good predictions despite an imperfect conceptual model
(i.e., the LSTM compensates for model structural error). This behavior is apparent in the variation of the recession constants
for Model 2 ($k$) and Model 3 ($k_1$ and $k_2$). In situations of low flow, the recession constants increase, whereas for situations of
high flow, the reverse is true.





**Over-parameterized constraint (Model 4)** In the case of an over-parameterized conceptual model, the role of the data-driven component is somewhat unclear. All parameters might be tweaked simultaneously in a manner that changes over time, to achieve a best-possible fit with the observed data. Such a case would presumably spoil any attempts to interpret the inner functioning of the hybrid model. However, in this case, we observe that the parameters of Model 4 (Fig. 2p) are optimized to
have almost constant values. In fact, the LSTM is able to correctly identify that the threshold parameter $S_0$ is not meaningful in predicting the output variable, so it is efficiently driven to a value of $0.0$. By doing so, the LSTM transforms the prescribed constraint in the form of the over-parameterized conceptual model into an architecture that is equivalent with the true one. This allows the LSTM to identify the true values of the other three parameters.

In Appendix B, we present four additional hybrid model versions that cover one under-parameterized case (Model 5 lacks
the parameter $\beta$), and three over-parameterized cases of different types (concerning model structure and parameters). The insights from these scenarios match what we have reported for the three broad classes above: the under-parameterized model struggles with the effect that parameters are heavily varied, while the LSTM in over-parameterized models produces almost static parameters in a combination that counteracts the over-parameterization best.

### 3.3.2 Measuring entropy of conceptual model parameter space

To quantify the variability of LSTM-predicted parameter values over time, we aggregate all individual values into a sample. These samples are shown as distributions in subplots g), j), m) and p) of Fig. 2. Wide distributions result for cases where parameters vary significantly over time, and very narrow distributions for cases of almost static behavior. We can quantify the entropy of the *joint* distribution of the parameters by using Eq. 6 as described in Sect. 2.3, with entropy being larger for wide distributions and lower for narrow distributions. Let it be noted that we are calculating the entropy of the parameters predicted
by the neural network which occupy a range of values from $0$ to $1$ as shown in the right-hand side of the right-most subplots in Fig. 2, such that the measurements of entropy are not affected by the scale of the parameters. Hence, these values occupy a range of values of 1, and considering that the maximum entropy (of a uniform distribution) over this value range is $0.0$, the calculated entropies are negative (with more negative meaning smaller entropy which equals smaller variability).

**Perfect physics constraint** Comparing the entropies obtained for Models 1, 2 and 3, we can confirm that Model 1 (LSTM
coupled to the true conceptual model) shows the lowest entropy. In a theoretical ideal case, the LSTM would have been able to perfectly recover the true values of $\alpha$, $\beta$ and $k$ without any variation in time at all, and that would lead to a theoretical entropy of $-\infty$ (but this is an unrealistic expectation given the difficult task required of the LSTM, and numerical imprecisions). Nevertheless, the variations of the parameters are very small and thus also the calculated entropy is significantly smaller than for Models 2 and 3.

**Imperfect physics constraint** We find that Model 2 generates less entropy than Model 3, which means that the conceptual model in Model 2 better represents the true model underlying the observed data (while definitely being further from the truth than Model 1). In this sense, the proposed entropy measure can be considered to represent "closeness" of a model's representation of the true system.





**Over-parameterized constraint** Measuring the entropy of the parameters for Model 4 distorts this result. As Model 4

permits a parameter configuration that makes the model equal to Model 1, the predicted parameters of the LSTM are again almost constant, and the calculated joint entropy is even lower than for Model 1. Note that this is a special case of an over-parametrized model. In Appendix B, and particularly in Fig. B1g we show an example of an over-parametrized model in which the true model is not findable.

**Comparing Conceptual Constraints on the Entropy Axis** To gain more intuition about how our hybrid models are ranked

based on entropy, we place them all (including the ones presented in Appendix B) on the same entropy axis (Fig. 3).

We would expect to find Model 1 furthest to the left in Fig. 3, because the LSTM has nothing to fix, so parameters are practically constant over time and their joint entropy is minimal. However, we see that is not the case and, among the models discussed in this section, it is Model 4 which creates the lowest entropy.

This is an artifact of comparing entropies in different dimensions. As an example, consider $X$ to be a random variable that

follows a multivariate Gaussian distribution, i.e. $X \sim \mathcal{N}(\mu, \Sigma)$ with $\mu \in \mathbb{R}^d$ and $\Sigma \in \mathbb{R}^{d \times d}$. The entropy of $X$ is then given as:

$$H(X) = \frac{d}{2}\log(2\pi e) + \frac{1}{2}\log(det(\Sigma)) \tag{7}$$

We can see that the entropy of $X$ is directly proportional to the determinant of $\Sigma$. If we add a single dimension to $\Sigma$ with a very low value on the main diagonal (a timeseries of almost-constant values will have close-to-zero variance) and all off-diagonal entries being practically zero, the value of entropy tends to decrease because the decrease of the second term through

multiplication of the original determinant with a value smaller than one tends to outweigh the increase of the first term.

There are two additional cases which show lower entropy due to the number of parameters in their conceptual models (Models 6 and 8) and one further example which has entropy close to Model 4 because it shares a similar inflow-outflow relationship (Model 7). The issues with these results are explained further in Appendix B. While explainable through theory, this ranking is counter-intuitive and does not meet our expectations for a metric that unifies the evaluation of arbitrary hybrid

models. We have illustrated these results here to allow the reader to follow our argument and move with us deeper into the hybrid models, i.e., into the LSTM hidden-state space.

### 3.3.3  Measuring entropy of LSTM hidden state space

To overcome the challenge of appropriately comparing the "activity" of the LSTM for models with differing numbers of parameters, we propose that the entropy of the coupled system should not be measured in the space of the parameters but in the

space of the hidden states of the LSTM instead. Because all of the networks in this example have the same number of hidden states (10) which move in the same range of values ($-1$ to $1$ due to the $tanh$ function in the operations in the network), the calculated entropies will be comparable between themselves.

The entropy values obtained for the hidden state spaces of all four models are reported in the left column of Fig. 2. The hidden states of the LSTM in Model 1 have smaller variations than in the rest of the models, and thus the entropy of this





network is the lowest among all candidates. This measure of variability has an even more intuitive interpretation as how much the LSTM has to compensate for a misspecified conceptual constraint.

**Comparing Conceptual Constraints on the Entropy Axis** When placing the models on our universal entropy axis in Fig. 4, Model 1 now appears furthest to the left, which meets our expectation that the true constraint should coincide with minimal "activity" of the LSTM. We also see the same ranking between Model 2 and Model 3, which again makes intuitive sense,

as using a one-reservoir-model better matches the true system. Finally, rearranging by the entropies of the LSTMs, Model 4 is now to the right of Model 1, which identifies it as a misspecified conceptual model, but honors that the resulting hybrid configuration is very close to the true system, as opposed to the proposed configurations in Models 2 and 3. Hence, measuring the entropy of the LSTM prevents us from disingenuous conclusions obtained by making unfair comparisons between models with different number of parameters.

**Pure LSTM as a Reference** One more advantage of measuring the entropy directly in the hidden states of the LSTM is that any hybrid model can now be compared to a single "pure" LSTM, i.e., an LSTM with a simple linear head layer instead of the conceptual model. The addition of a conceptual head layer should make the prediction task of the LSTM easier - at least this is the prevailing idea when promoting "physics-informed" ML. In our setting, adding useful information through the conceptual constraint should reduce the required activity of the LSTM, and hence, entropy. If, by contrast, the conceptual constraint made

the task even more difficult, it would add entropy. Marking the pure LSTM as Model 0, we can create a divide on our axis between models that add "good" (helpful) physics (here: Models 1 and 4), and models which add "bad" (misleading) physics (here: Models 2 and 3). In addition, Models 5, 6, 7 and 8 are discussed in Appendix B, where it is shown that they also fall consistently in these categories of "good" and "bad".

**On the Complexity of the Prediction Task** The LSTM by itself can be seen as a baseline of the required complexity for

accomplishing the prediction task. The proposed measure of entropy can be related to the overall complexity of the network as the entropy of the trajectories of the states in dynamical systems has been related to their Kolgomorov complexity (Galatolo et al., 2010). In theory, if the true model is specified as the conceptual head layer, the entropy of the LSTM is reduced to the theoretical minimum ($-\infty$) and the required entropy or complexity to accomplish the specific modeling task is completely contributed by the conceptual head layer. Hence, the entropy of the conceptual head layer in Model 1 should be exactly the

same as the entropy of the pure LSTM (Model 0), but measuring the entropy of the conceptual head layer by itself is not straight-forward and remains an open challenge.

## 3.4 Summary of the proposed approach

Let us distill our proposed approach as a diagnostic framework that discerns the adequacy of conceptual constraints in hybrid models. When the prescribed conceptual model accurately represents the natural system, the LSTM will exhibit minimal inter-

vention, effectively endorsing the conceptual model. Conversely, when the conceptual constraint fundamentally misrepresents the system dynamics, the LSTM will demonstrate high activity, working extensively to overcome the inherent limitations of the prescribed conceptual model. This difference in LSTM activity serves as a clear signal for assessing the fidelity of our initial conceptual model





These situations can be detected by the following proposed workflow:

1. Visualize the time-sequence of LSTM-predicted parameters to gain insight about how heavily the data-driven component acts against the physics constraint; draw conclusions about compensation mechanisms and judge whether the physics constraint is sufficiently honored or massively altered in the hybrid model.

2. Quantify the joint entropy of the LSTM hidden state space trajectories; compare against a pure LSTM for the prediction task for reference, and, ideally, against alternative formulations of conceptual constraints by placing all resulting

entropies on the universal model evaluation axis.

3. Interpret the results: are the conceptual components of the hybrid models an advantage or a burden in solving the prediction task? Which configurations are more helpful than others? Try to understand why from step 1. Over-parameterization will tend to be helpful but with some parameters driven to "unphysical" values; under-parameterization will make the task unnecessarily difficult.

From the analysis of the didactic examples, we specifically want to highlight that the constraint-morphing capability of the data-driven component is both an opportunity and a risk: it is very promising to see that the flexibility of the LSTM is not abused, but rather it points us towards parsimonious model structures (as in Model 4). At the same time, this constraint-morphing happens under the hood (e.g., resulting NSE is practically the same for *all* our analyzed model versions!) - *it is not safe to say that a hybrid model naturally satisfies the constraint we have prescribed*. As such, we should be careful with stating

that a model is "physics-constrained" before investigating in detail what the final version of the LSTM is doing. This is where our proposed diagnostic routine helps.

One last point is that the issue of uncertainty is not addressed by these examples because the output of the true model and therefore our observed data were unaffected by noise. Our measurement of entropy can definitely be a part of a larger and more comprehensive framework which includes both epistemic and aleatory uncertainty (Gong et al., 2013) and probabilistic model

representations, but such a framework is beyond the scope of this article. Nevertheless, in any measurement of entropy there will always be a fraction given by the intrinsic chaos of data.

## 4   Case study: CAMELS-GB

Following the intuition developed by the didactic examples, we apply our developed metric to a case study in large sample hydrology using the CAMELS-GB dataset.

Both the pure LSTM and the LSTMs coupled with the conceptual models have 64 hidden states each, which makes them directly comparable between themselves. All models were trained using the Adam optimizer (Kingma and Ba, 2017) with a learning rate of $1 \times 10^{-3}$ and a different number of epochs depending on the model, with the number of epochs always ranging between 28 and 32. The ranges allowed for the parameters of the conceptual models are listed in Table A2, the static attributes used as input to the LSTM in all models are listed in Table A3.





Further details about the study setup are presented in Appendix A2 but this analysis follows the results from Acuña Espinoza et al. (2024), so we first summarize their main findings to put these new results into context. The meticulous reader will notice some differences in the results between the previous study and these current results. These differences are discussed in Appendix C and do not impact the main findings in either study.

## 4.1    Motivation or: why do we want hybrid models?

Hybrid models have demonstrated significant improvements in hydrological predictions across multiple applications. These include enhanced accuracy in daily streamflow prediction (Jiang et al., 2020; Feng et al., 2022), better predictions in large basins (Bindas et al., 2024), and more precise estimates of variables like volumetric water content (Bandai et al., 2024) and stream water temperature (Rahmani et al., 2023), as some examples. In each case, the hybrid approach outperformed traditional physics-based conceptual models, including the EXP-Hydro and HBV models, the Muskingum-Cunge river routing method,

and a partial-differential-equation-based description of the physical process, respectively. However, while these improvements are of note and leaving aside aspects of lacking interpretability, the central question of "to bucket or not to bucket" was: given the remarkable success of purely data-driven approaches, is the additional effort of combining them with conceptual models actually worth it?

        Acuña Espinoza et al. (2024) conducted a model comparison study that evaluated four different approaches: a purely data-

driven LSTM and three conceptual hydrological models, each later transformed into hybrids through the process described in Section 2.2.2. The three conceptual models: SHM, adapted from Ehret et al. (2020), Bucket, and Nonsense represent different hypotheses of the hydrological system. Among these, SHM is a conventional hydrological model suitable for practical applications, while Bucket and Nonsense serve as contrasting cases: Bucket being an oversimplified representation and Nonsense incorporating physically implausible assumptions.

We evaluate the streamflow prediction performance of these seven models using the Cumulative Density Function (CDF) to aggregate model performance across all 671 basins in the dataset. Figure 5 presents these results, while Table 1 provides key metrics derived from the CDF analysis. The two considered metrics are the median NSE, which corresponds to the CDF's middle quantile (the higher the better), and the "area under the curve" (AUC). The AUC serves as a summary metric where lower values indicate better performance, because the AUC becomes minimal if NSE only takes on maximum values (Gauch

et al., 2021).

        Figure 5 demonstrates the effect of combining conceptual models with LSTM networks. The effect is visible as a drastic shift to the right from the dashed lines (purely conceptual models) to the solid lines (their hybrid counterparts). This improvement is further quantified in Table 1, where the metrics consistently show improved performance for hybrid versions compared to their original counterparts.

Despite these improvements, our results show that the incorporation of the conceptual models did not exceed the performance of a pure LSTM approach (in Figure 5, the LSTM is farthest to the right). Interestingly, performance improves the most by hybridizing the oversimplified Bucket model *and* it is the hybrid model that matches the LSTM most closely. Intuitively, one might have expected that the flexibility of the LSTM helps the Nonsense model the most, followed by the Bucket model and




**Table 1.** Comparison of model performance quantified by area under the NSE curve (AUC) and median NSE

| Model | AUC | Median NSE |
|---|---|---|
| LSTM | 0.123 | 0.865 |
| SHM | 0.267 | 0.747 |
| Hybrid SHM | 0.216 | 0.839 |
| Bucket | 0.395 | 0.582 |
| Hybrid Bucket | 0.147 | 0.852 |
| Nonsense | 0.477 | 0.511 |
| Hybrid Nonsense | 0.265 | 0.801 |

finally the SHM model; and that after hybridization, the performance of Hybrid SHM is best and better than that of the pure
LSTM. What we observe instead suggests that the SHM constraint actually limits the hybrid performance, adding a Bucket-type
constraint is more successful, and none of these constraints are helpful in improving prediction skill.

These findings raise several urgent questions:

– Why do apparently "bad" physics allow for better hybrid performance than "good" physics?

– What can we conclude from hybrid performance after all if it does not reflect process fidelity?

– Do physics get in the way of successful data-driven modeling?

We note that one untouched advantage of the hybrid approach lies in its ability to directly derive unobserved variables, such
as soil water equivalent (SWE), without requiring secondary models. Hence, we wish to provide modelers with tools to obtain
satisfying answers to these questions and to better inform and justify hybrid modeling in future research and practice.

## 4.2 Performance on individual basins

To better understand the mechanisms of these hybrid models and their impact on model performance, we will investigate the
prediction task for five individual basins in detail. These five basins were carefully chosen to facilitate discussion in this section.
In Sect. 4.3.3 we draw statistical conclusions about the prevailing behaviors for all basins.

Figure 6 exemplarily shows five basins where the performance gap between hybrid and non-hybrid versions is again very
clear; however, these basins all share the characteristic that all hybrid models, including the deliberately implausible Nonsense
model, reach very similar performance. This seems counterintuitive in several aspects and again supports the research questions
we have formulated above: We would have expected to see a difference in the performance of the hybrid models, depending on
the constraints imposed. Does the LSTM really not care, that is, is any constraint simply transformed into the same effective
end product? Further, we would have expected (hoped?) that at least the physics-plausible constraint of the SHM actually helps
solve the prediction task; this is only observed in one of the five basins; yet, confusingly enough, in that specific case, *all* of the
constraints (physics-plausible or not) seem to help. Overall, Figure 6 highlights the urgent need for diagnostic analysis tools





that help us find out what it actually means to constrain a data-driven model with a conceptual hydrological model and how much physics remain inside.

Since we are in a real-data setup now, there is no "true" model or constraint that we could use as a reference for minimal entropy on our evaluation axis. We will therefore seek the LSTM component that produces the least entropy. Our main anchor
will be the performance of the pure LSTM, dividing between meaningful added knowledge and misguided assumptions that require compensation by the LSTM.

### 4.3 Analysis and interpretation of results

#### 4.3.1 Measuring entropy of LSTM hidden state space

Following the intuition developed in Section 3, we address the questions in the previous section through an entropy analysis of
the LSTM's hidden states for the prediction of the five individual basins introduced above.

Figure 7 shows the calculated entropy during the testing period for both pure LSTM and hybrid models. This is equivalent to the entropy axis we had introduced in our didactic examples, with the pure LSTM marking the divide between "good" and "bad" constraints. Overall, we find that the ranking varies per basin: in some cases (basins 23008, 18014, 41025), the pure LSTM shows by far the lowest entropy and hence none of the constraints can be considered useful for predicting streamflow
at these basins; for the other basins, at least some conceptual constraints proved helpful, in basin 5003 even *all* of them.

Focusing on basin 5003, the observed ranking aligns with our expectation that SHM is the only plausible and hence most useful constraint. This suggests that SHM's imposed structure reasonably reflects the natural system, effectively transferring part of the entropy to the physics-based component. However, *any* conceptual hydrological model reduces the network's entropy compared to the pure LSTM, even the Nonsense model, which opposes our expectation that this constraint should not be
useful. Notably, Hybrid Nonsense shows even lower entropy than Hybrid Bucket, indicating that some complexity is necessary and a too simple conceptual model offers little benefit.

Basin 73014 presents a counterexample where Hybrid Nonsense performs best (produces the least entropy) and Hybrid SHM is located to the right of the LSTM, suggesting that a plausible hydrological model can even make predictions more difficult. This finding highlights the unpredictability of hybrid models and the need for deeper investigations to achieve interpretability;
simply imposing a constraint does not do the job.

#### 4.3.2 Visualization of time-varying parameters

In this section, we demonstrate the power of visually analyzing the time-series of the LSTM-predicted parameters on the example of exploring why the Nonsense model creates the least entropy for basin 73014. Figure 8 (top panel) compares the differences between observed and simulated streamflow values for the non-hybrid and hybrid versions of the Nonsense model in
this basin. The Hybrid Nonsense model shows drastically improved predictions, represented by the solid line in the streamflow plot, compared to its non-hybrid counterpart (dashed line). That means that allowing the parameters to vary over time greatly improves the ability of this model to make accurate predictions.





To better understand the adjustments made by the LSTM, let us first look at the original structure of the Nonsense model, which is considered physically implausible due to the arrangement of its hydrological storage units (see schematic illustration in Figure 1c). Counter-intuitively, water from direct precipitation or snowmelt initially enters the model through the baseflow storage, typically considered the unit with the longest retention time. The model then routes water through an unusual sequence: it moves next to the interflow storage which once again has a longer residence time, then it passes into the unsaturated zone, loses some mass through evapotranspiration, and is finally transformed into the streamflow output. Ignoring their physical interpretation for a moment, the Nonsense model basically consists of a series of three storages connected sequentially, forming a cascade-like arrangement that essentially transforms the storages into a dampening function, which delays the input signal.

The adjustments made to this implausible model structure by the LSTM component of Hybrid Nonsense become apparent when examining its states and parameters (bottom left and right panel in Figure 8). To simplify interpretation, all plots have been normalized by the mean value of the corresponding state or parameter over the analyzed period. This normalization sets the mean value to 1.0 on the plot, with the lines indicating deviations from the mean. However, that doesn't mean that parameters for Nonsense and Hybrid Nonsense have similar or even values that are close to each other. For example, the value of $s_{u,max}$ for Nonsense is 398.6 mm while the mean value for Hybrid Nonsense is 83.5 mm. This reflects a typical behaviour in these hybrid models were the the parameters for the non-hybrid and hybrid model can occupy drastically different ranges.

Analyzing the static parameters in the Nonsense model, the dampening behavior of the storages becomes evident: the dashed lines for the baseflow storage *sb* and the interfow storage *si* closely resemble the output hydrograph, but are dampened too strongly in the unsaturated zone storage *su*. However, this behavior changes significantly when the model becomes Hybrid Nonsense. Specifically, the line for *sb* becomes horizontal, indicating that the parameter $k_b$ is being used to effectively "skip" this storage.

Additionally, the solid lines for *su* and $s_{u,max}$ reveal a distinct pattern in which $s_{u,max}$ closely tracks the value of *su*. This behavior is tied to the conditional property of the storage: if $s_u > s_{u,max}$, any excess runoff added to *su* is immediately outputted. Because $s_{u,max}$ consistently mirrors *su*, any additional runoff into this storage is immediately converted to simulated streamflow, once again, effectively bypassing this storage. In addition, the outflow of *su* is also managed by $\beta$ which appears to be anticorrelated with *si* / $k_i$ to match the shape of the observed hydrograph. As a result, the Hybrid Nonsense model essentially functions as a single-storage system with added lagging behavior. This lag is introduced by the sequential transfer of mass between the storages, which occurs one at a time during each time step.

The imposed structure of the original Nonsense model was effectively modified by the LSTM, transforming the overcomplicated but physically implausible model into something that more closely resembles the Bucket model, with some additional flexibility guided by the characteristics of the training data. Since we did not impose specific constraints on the storage behavior, apart from limits to the parameters, the LSTM discovered an optimized architecture that, in combination with the data-driven component, works just as well as any of the other constraints. It seems that the modified Nonsense structure is significantly more suitable than the oversimplified Bucket model, presumably because it allows for just the right amount of additional freedom. Interestingly, morphing the structure of the Nonsense constraint costs the LSTM less effort (entropy) than fighting against (arguably) more adequate but too rigid constraints such as the SHM or the Bucket conceptual models - this





is important to keep in mind when interpreting the results of our entropy analysis. High entropy clearly indicates struggling caused by the imposed constraint; low entropy paired with unaffected parameters means a plausible constraint, whereas low

entropy paired with suspicious time-varying patterns that alter the qualitative behavior of the states means overwriting of constraints in favor of something more efficient; something that can potentially still be meaningful, as we have uncovered here, and also from the over-parameterization cases in our didactic examples (so, there is hope).

Figure 9 presents the same analysis period for the SHM and Hybrid SHM models in basin 73014. Due to its larger structure, interpretation becomes more challenging, but we observe some of the same behaviors identified in the analysis of Hybrid

Nonsense. The LSTM determines that some of the additional storage compartments in SHM are unnecessary, as it does not utilize *sb* and instead regulates outflow through the fast-flow (*sf*) and *si* storages. Furthermore, minimal changes in *si* suggest that most of the outflow is directed through *sf*.

The high variability in the LSTM-controlled parameters for the remaining reservoirs may show a learned behavior from other basins in the dataset. Such adaptation appears unnecessary for this particular basin, as the predictions made by the non-

Hybrid SHM model were already sufficiently accurate and modifications made by the LSTM improved performance only slightly. Ultimately, this leads to the Hybrid SHM model being penalized, placing it last in our ranking. As a final note, that the "intervention" of the LSTM was not obvious from comparing the hydrographs produced by SHM and Hybrid SHM and without the analysis proposed here, one would think that Hybrid SHM is a well-constrained hybrid model that respects the assumptions formulated in SHM - which is not at all the case, as we have shown here.

### 580   4.3.3   Statistical analysis of results for all basins

The most common pattern across the other basins 23008, 18014, and 41025 with the LSTM consistently having the lowest entropy shows a non-consistent ranking of the hybrid models. It seems their ranking is determined by the specific hydrological system to be modeled and the required model complexity. We therefore analyze here the overall statistics and rankings of entropy across all basins.

The violin plots in Figure 10 show the entropy distributions of each model, with median values of $-151.2$ nats for LSTM, $-128.2$ nats for Hybrid Nonsense, $-113.1$ nats for Hybrid SHM, and $-111.0$ nats for Hybrid Bucket. The LSTM's wider dispersion highlights the varying complexity required to model each individual basin. The fact that the LSTM is the only one to cover much of the lower entropy range demonstrates that, in most cases, introducing a conceptual constraint substantially increases the modeling challenge.

To analyze this in more detail, we collected the individual rankings per basin for the whole dataset, identified the unique rankings that appear, and determined their frequency of occurrence. The counts of rankings are shown in Figure 11.

The ranking suggested by the medians in Figure 10 (entropy of LSTM being lowest, followed by Hybrid Nonsense, Hybrid SHM, and Hybrid Bucket) reflects most frequent ranking across all basins (67%). Aggregating all those basins, for which the pure LSTM obtains the lowest entropy, leads to 91% of all basins. This tells us that, in general over this particular large-sample

dataset, the conceptual representations used in our hybrid models were not able to make the prediction task easier for the LSTM and the prior knowledge that we tried to enforce didn't help. Ultimately we got a hybrid model that predicted well not because





of the physical constraints that we imposed, but because the LSTM was able to compensate for these constraints through added effort (entropy).

Although our previous statement is the main finding of this study, we are still able to identify specific catchments for
which the added prior knowledge indeed helped. In Figure 11 there are 11 basins (1.6%) which show the LSTM at the top, meaning that any of the conceptual models added information that helped in prediction; in 8 out of 671 cases (1.2%), Hybrid SHM showed the lowest entropy, meaning that the constraint that a hydrologist would perceive as the most plausible and useful actually turned out to need the least compensation by the data-driven component. And only in 1 out of 671 cases, the physically least plausible Nonsense model needed the most compensation.

These results overwhelmingly suggest that we need to reconsider our ways of building hybrid models (and even conceptual models). Even if "just" the parameters of conceptual hydrological models are modified by the data-driven component, the resulting hybrid models function differently than what we expect by imposing mass balance equations.

### 4.4 What about other approaches to hybrid modeling?

Measuring the entropy of the data-driven component of a hybrid model works particularly well in our setup because of the
tight coupling between the LSTM and the conceptual hydrological model through the parameters of the latter. Nevertheless, our suggested approach can be effectively applied to other types of hybrid models or physics-informed machine learning.

As one example of an alternative approach to physics-informed machine learning, post-processing the results of a hydrological model to improve its predictions has been suggested (Nearing et al., 2020; Frame et al., 2021). For this application, a traditional hydrological model with static parameters makes an initial run to predict streamflow, and the predictions as well
as the states of the hydrological model are fed to an LSTM to make improved predictions of streamflow. The approach is successful in the sense that it improves predictions of streamflow and manages to move all performance CDF curves close, but not beyond, the LSTM as shown in Figure 12. Note that both the LSTM and post-processing LSTMs use the same number of hidden nodes (64), making our approach and comparison still applicable.

The violin plot of entropies for the LSTM hidden states across all basins shown in Figure 13 suggests a different conclusion
than for our previous hybrids. It seems that the LSTM is largely "unimpressed" by the additional input, no matter which model it was created by; only the Nonsense model (of all things!) seems to be able to effectively reduce the effort required for the prediction task, meaning that there is some pre-processing that this particular model does that is actually helpful. Figure 14 shows a much more mixed bag of results where, for certain specific basins, any of the conceptual models might produce an output that reduces the entropy of the LSTM. Considering that feeding the model "Nonsense" is helpful in close to 80%
of all basins should again be an impressive warning that feeding physics-based model output to a data-driven model is not necessarily physically meaningful (in that case, we would expect the LSTM to have a harder time with nonsense outputs). This confirms previous findings in literature which suggest that post-processing a conceptual model is not a good method to make "physics-informed" machine learning (Nearing et al., 2020; Frame et al., 2021).

Since post-processing conceptual models do not allow for scrutinizing conceptual states or parameters, as with hybrid mod-
els, we cannot perform our detailed analysis as shown above (Figures 8 and 9). Hence, interpretation of the impact of the





physics-based input (hard to call this a constraint) requires interpretation of the LSTM hidden states. This is a current line of research in its own right (e.g., Feng et al., 2024; Blougouras et al., 2024), and goes beyond the scope of this study. It will be interesting to explore what the contribution of "Nonsense" is that seems to simplify the prediction task for the LSTM, while physically-meaningful outputs as LSTM inputs do not necessarily help.

## 4.5 Summary of findings from the case study

In this case study, we compared a pure LSTM model with three hybrid hydrological models based on the CAMELS-GB large-sample data set. Overall, we found that the LSTM outperformed the hybrid models in predicting streamflow, and our entropy analysis revealed that adding physics-based constraints generally did not simplify the prediction task.

Our analysis also showed that the LSTM effectively adjusts the constraints imposed by the conceptual model. For instance, Hybrid Nonsense is very different from its original Nonsense formulation with the LSTM identifying an optimized architecture that, when combined with a data-driven component, performs just as well as all other models. The degree of effort required for the LSTM to modify these constraints provides insight into how accurately the conceptual model represents the underlying system. This finding highlights a key opportunity for hybrid modeling: refining existing models to better suit specific sites based on training data characteristics. In essence, hybrid models can guide us toward more parsimonious model structures.

Notably, this process occurs entirely under the hood. If we had evaluated performance using only NSE, we might have mistakenly concluded that the Nonsense constraint was just as valid as SHM or Bucket, since all three achieved the same performance when paired with the LSTM. These results overwhelmingly suggest that we need to reconsider our ways of building hybrid models. Even if "just" the parameters of conceptual hydrological models are modified by the data-driven component, the resulting hybrid models function differently than what we expect by imposing mass balance equations.

Finally, we have provided an outlook of how to apply our entropy-based analysis to other types of hybrid construction. While the in-depth analysis of resulting architectures relies on the existence of conceptual states and/or parameters in the hybrid model, the evaluation of entropy in the LSTM hidden states already provides an objective insight that would have been obscured when only considering skill scores, as pointed out above. Complementary analyses that target the specifics of other hybrid model architectures are left for future work.

## 5 Conclusions

"Man is always prey to his truths. Once he has admitted them, he cannot free himself from them" (Camus, 1991). The pursuit of a single, universal model to explain every hydrological system is fundamentally absurd. This paper challenges the hydrological community's tendency to rely on a single model, like SHM, as a comprehensive explanation for the complex dynamics of all river basins. Our work shows that SHM, or other conceptual models of its kind, is precisely the kind of rigid "truth" Camus warned us about: a single model that, now coupled with a component that learns from data, represents a seemingly straightforward explanation of every and any complex natural system.



The recognition of these limitations extends to the process of model selection itself. As observed by Kuhn and Hacking (2012), "scientists work from models acquired through education and subsequent exposure to the literature, often without fully knowing what characteristics have elevated these models to the status of community paradigms." This implicit acceptance of certain modeling approaches, while pragmatic, further highlights the tension between using an interpretable model and capturing the full complexity of real-world systems. Hybrid models acknowledge this tension, incorporating prior knowledge to achieve partial interpretability while accepting the residual complexity that remains unmodeled. However, our study suggests that this balance is often skewed in favor of the data-driven component. The use of a conceptual model as a structural prior is an attempt to extract meaningful dynamics from a larger environmental system (Young et al., 1996), but as we have shown, this attempt is often forced. When the problem is relatively simple, such as predicting streamflow in Sect. 4, conceptual prior knowledge is largely ignored in favor of a more flexible, data-driven structure, raising the question of whether it was necessary in the first place.

Interestingly, measuring the "complexity" of the LSTM in the sense of measuring its "compensation activity" when coupled to a parsimonious but inadequate representation yields larger entropy than when coupled to a more flexible inadequate representation. This means two things: low entropy does *not* necessarily mean that the constraint is honored and that it is realistic; and: the LSTM seems to have more work to do to fight against something simple but wrong, than to fiddle around with an arbitrary, flexible-enough structure to make it work. We have seen this in the over-parameterization cases in the didactic examples and with the Nonsense model in the case study. So, high entropy means the LSTM is struggling because of a too rigid, inadequate constraint. Low entropy means that the LSTM is seeing something in the constraint; however, deeper analysis in the form of inspecting parameter and state trajectories are required to distinguish whether the constraint is deemed reasonable to predict the data (constant parameters), or whether it just lends itself well to be transformed into something new, parsimonious, and effective, which might even be physics-explainable and guide us towards a better representation of the true model (so, there is hope in hybrids; just in a different way than the community might have anticipated).

Our primary contribution is a metric that quantifies how much prior knowledge contributes compared to a purely data-driven approach. In Sect. 3, we demonstrated how different conceptual models can be evaluated based on how closely their prescribed equations align with those governing the "true" system. Additionally, we showed that a data-driven model can serve as a reference point, distinguishing between conceptual models that better approximate reality and those that do not. We propose that data-driven models should serve as the baseline for evaluating hybrid models, which allows us to determine whether incorporating prior knowledge, such as physics-based constraints, genuinely enhances predictive performance or simply adds unnecessary complexity.

Applying this metric to a large-scale case study revealed that our attempts to improve predictive capacity through hybridization were often unsuccessful. This was because the added knowledge rarely captured the true system dynamics, forcing the flexible component of our models to compensate for incorrect assumptions (Sect. 4.3.2). Nevertheless, these results suggest a promising path forward: in the future, hybrid models could become valuable tools for refining our understanding of hydrological systems, but only if we critically reassess traditional modeling practices. The fact that even a "nonsense" conceptual model demonstrated the highest potential for adding useful information in post-processing hybrids raises new questions. We





hypothesize that physics constraints in the form of strict sequential processing may be too rigid and that guiding the LSTM toward learning appropriate lag functions could be a more effective strategy.

Overall, our findings challenge the assumption that physics-informed machine learning necessarily preserves the physics as initially formulated. Instead, the data-driven component may restructure the imposed framework, uncovering a more effective, potentially physics-explainable alternative. We do not oppose hybrid modeling; rather, we propose a quantitative tool to analyze how much the physics-based constraints are modified and suggest a workflow for diagnosing these structural adaptations. In the end, hybrid modeling, when paired with information-theoretic analyses, has the potential to provide valuable physical insights. Without such an approach, however, many so-called "physics-informed" models may be better described as physics-ignored.

*Code availability.* The code to recreate any of the experiments in this paper is publicly available at https://github.com/manuel-alvarez-chaves/ hybrid-models/tree/paper (last access: 25.03.2025). We also used part of the Hy2DL library which can be accessed as a brach at https://github. com/KIT-HYD/Hy2DL/tree/manuel-dev (last access: 25.03.2025). The UNITE toolbox is also available at https://github.com/manuel-alvarez-chaves/ unite_toolbox (last access: 25.03.2025).

*Data availability.* CAMELS-GB is available at https://doi.org/10.5285/8344e4f3-d2ea-44f5-8afa-86d2987543a9. All of the code for this project, model state dictionaries, model configurations, training logs and netCDF files of the results of this study have been archived at the data repository of the University of Stuttgart (DaRUS) and can be found at this link: https://doi.org/10.18419/DARUS-4920

## Appendix A: Study setup

We provide further details of our experimental setup in this section, and point the interested reader towards a repository in the data repository of the University of Stuttgart (DaRUS) that contains all the code for this project including: training scripts, logs and archived netCDF files of the saved internal and hidden states, fluxes and predictions made by each model. The repository can be found at this link: https://doi.org/10.18419/DARUS-4920.

### A1   Didactic examples

The ranges for the parameters of the conceptual models allowed in the didactic examples are listed in Table A1.

### A2   CAMELS-GB

The ranges for the parameters of the conceptual models used in the CAMELS-GB case are listed in Table A2. The static attributes used as input to the LSTM in all models are listed in Table A3.





**Table A1.** Ranges of the model parameters in the synthetic examples

| Model | Parameter | Limits | |
|---|---|---|---|
| | | *Lower* | *Upper* |
| | $\alpha$ | 0.5 | 2.0 |
| 1 | $\beta$ | 0.1 | 2.0 |
| | $k$ | 10.0 | 60.0 |
| | $\alpha$ | 0.5 | 2.0 |
| 2 | $s_{max}$ | 50.0 | 400.0 |
| | $k$ | 1.0 | 100.0 |
| | $\alpha$ | 0.1 | 2.0 |
| 3 | $k_1$ | 30.0 | 300.0 |
| | $k_2$ | 0.1 | 40.0 |
| | $\alpha$ | 0.5 | 2.0 |
| | $\beta$ | 0.9 | 3.5 |
| 4 | $s_0$ | 0.0 | 50.0 |
| | $k$ | 1.0 | 60.0 |
| | $\alpha$ | 0.5 | 2.0 |
| 5 | $k$ | 0.1 | 300.0 |
| | $\alpha$ | 0.5 | 2.0 |
| | $\beta$ | 0.9 | 3.5 |
| 6 | $\gamma$ | 0.5 | 2.0 |
| | $k$ | 1.0 | 60.0 |
| | $\alpha$ | 0.1 | 2.0 |
| | $s_{1,max}$ | 1.0 | 200.0 |
| 7 | $k_1$ | 100.0 | 400.0 |
| | $\beta$ | 0.1 | 2.0 |
| | $k_2$ | 1.0 | 60.0 |
| | $\alpha$ | 0.5 | 2.0 |
| | $s_{max}$ | 1.0 | 250.0 |
| 8 | $\beta$ | 0.9 | 3.5 |
| | $k$ | 1.0 | 100.0 |

## Appendix B: Additional didactic examples

In Section 3 and specifically Figure 2 we showed four key didactic cases that can be used to understand the rankings in Figures
3 and 4. To further our understanding of the behaviors that can be quantified by measuring the entropy of the hidden states



**Table A2.** Ranges of the model parameters for the CAMELS-GB case study

| Model | Parameter | Limits | |
|---|---|---|---|
| | | *Lower* | *Upper* |
| SHM | dd | 0.0 | 10.0 |
| | f_thr | 10.0 | 60.0 |
| | $s_{u,max}$ | 20.0 | 700.0 |
| | $\beta$ | 1.0 | 6.0 |
| | perc | 0.0 | 1.0 |
| | $k_f$ | 0.05 | 0.9 |
| | $k_i$ | 0.01 | 0.5 |
| | $k_b$ | 0.001 | 0.2 |
| Bucket | $k$ | 0.002 | 1.0 |
| | aux_ET | 0.01 | 1.5 |
| Nonsense | dd | 0.0 | 10.0 |
| | $s_{u,max}$ | 20.0 | 700.0 |
| | $\beta$ | 1.0 | 6.0 |
| | $k_i$ | 0.01 | 0.5 |
| | $k_b$ | 0.001 | 0.2 |

of the LSTM, we show an additional five examples in Figure B1. In this section we briefly describe what learned from these additional examples.

Model 0 represents a case where we only have an LSTM which predicts streamflow, i.e. a purely data-driven model. In this particular case we see that the pure LSTM is also able to make accurate predictions that are on par with the "true" model and all other alternatives. As such, this model serves as our baseline and any additional knowledge should make prediction easier

(reduce entropy) or more difficult (increase entropy).

Model 5 represents a case in which the added knowledge is lacking the degrees of freedom that the "true" model has. Thus, the LSTM has to take over and compensate using its internal hidden states, resulting in the high measurement of entropy of the LSTM. Although this behavior is also apparent in the variations of the parameters shown in Figure B1e, measuring entropy there can result in the wrong assumption that a conceptual model with more reservoirs in Model 3 would more closely resemble

the "true" model, as shown in Figure 3 but this is not true. The true picture is given by the entropy of the LSTMs in Figure 4.

Models 6 to 8 all serve as cases where the conceptual model has a greater number of parameters than the "true" model, making them overparametrized but cases such as 6 and 8 resemble the "true" model very closely. Model 7 also has the "true" model embedded within, but the input relationship is distorted by the extra reservoir in $s_1$.

Returning to the idea of how much a model closely matches the "true" model, the ranking in Figure 4 makes intuitive sense.

The "true" model is furthest to the left followed by Models 4, 6 and 8 which use the same number of reservoirs and whose



**Table A3.** Catchment attributes from the CAMELS-GB dataset used to train all models

| Type | Attribute | Description |
| --- | --- | --- |
| Topographic | area | catchment area (km$^2$) |
| Topographic | elev_mean | mean elevation (m. a. s. l.) |
| Topographic | dpsbar | slope of the catchment mean drainage path (m km$^{-1}$) |
| Soil | sand_perc | percent sand (%) |
| Soil | silt_perc | percent silt (%) |
| Soil | clay_perc | percent clay (%) |
| Soil | porosity_hypres | soil porosity calculate using the hypres pedotransfer function (-) |
| Soil | conductivity_hypres | hydraulic conductiviyu calculated using the hypres pedotransfer function (-) |
| Soil | soil_depth_pelletier | depth to bedrock (m) |
| Land cover | dwood_perc | fraction of precipitation falling as snow (for days colder than 0°C) |
| Land cover | ewood_perc | percent of catchment that is deciduous woodland (%) |
| Land cover | crop_perc | percent of catchment that is evergreen woodland (%) |
| Land cover | urban_perc | percent of catchment that is cropland (%) |
| Human influence | reservoir_cap | percent of catchment that is urban area (%) |
| Climatic | p_mean | catchment reservoir capacity (ML) |
| Climatic | pet_mean | mean daily precipitation (mm d$^{-1}$) |
| Climatic | p_seasonality | mean daily PET (mm d$^{-1}$) |
| Climatic | frac_snow | seasonality and timing of precipitation (estimated using sine curves) |
| Climatic | high_prec_freq | frequency of high-precipitation days ($\geq$5× mean daily precipitation) |
| Climatic | low_prec_freq | frequency of dry days (<1 mm d$^{-1}$) |
| Climatic | high_prec_dur | average duration of high-precipitation events ($\geq$5× mean daily precipitation) |
| Climatic | low_prec_dur | average duration of dry periods (number of consecutive days <1 mm d$^-$1) |

input-output relationships can match that of the "true" model (all have an exponential term). This grouping is followed by Model 7 which has the ability to match the output relationship using the second exponential reservoir but the input is not directly precipitation and evapotranspiration but some dampened product coming from the first reservoir. Next we have the divider between encoding "good" and "bad" physics. All of the previous models have the "true" model (in some sense) within

their structure. This additional knowledge makes the task required of the LSTM easier, thus reducing its entropy. If, instead, we encode "bad" physics, we fall in cases where the model is able to perfectly fit the observed data but not because of the additional knowledge, but because of a more complex LSTM which now, in addition to prediction, has to overwrite our incorrect prior knowledge.

Model 3 is the worst offender as the usage of a two reservoirs system with none of the reservoirs having an exponential term

in their output relationship is the case most dissimilar to the "true" model. Models 2 and 5 improve upon this condition, but still lack some semblance of the "true" model in their structure making them to also fall in the "bad" physics category.



**Appendix C: Standardizing training pipelines**

The results in the case study presented in Section 4 follow the results of the previous study of Acuña Espinoza et al. (2024).
However the data and metrics reported between studies related to model variables and performance are not the same. Between
studies we modified our training pipelines to adopt current standard practices (Shen et al., 2023; Kratzert et al., 2024), therefore
there are differences between the metrics.

For the models with static parameters, these differences are shown in Figure C1 and Table C1. In our previous study, the
conceptual models were calibrated individually for each basin using the DREAM algorithm (Vrugt, 2016). This procedure
results in better performance at predicting streamflow than in regional training as we did for this current study. The drop in
performance could be the due to the identification of regional sets of parameters, as in Tsai et al. (2021), but we did not pursue
this finding further.

**Table C1.** Comparison of model performance for models with static parameters quantified by area under the NSE curve (AUC) and median
NSE

| Model | Version | AUC | Median NSE |
|---|---|---|---|
| SHM | TBONTB | 0.243 | 0.760 |
| | New | 0.267 | 0.747 |
| Bucket | TBONTB | 0.381 | 0.590 |
| | New | 0.395 | 0.582 |
| Nonsense | TBONTB | 0.441 | 0.510 |
| | New | 0.477 | 0.511 |

Then for the models with dynamic parameters, the differences are shown in Figure C2 and Table C2.

**Table C2.** Comparison of model performance quantified by area under the NSE curve (AUC) and median NSE

| Model | Version | AUC | Median NSE |
|---|---|---|---|
| LSTM | TBONTB | 0.120 | 0.870 |
| | New | 0.123 | 0.865 |
| Hybrid SHM | TBONTB | 0.216 | 0.844 |
| | New | 0.216 | 0.839 |
| Hybrid Bucket | TBONTB | 0.168 | 0.857 |
| | New | 0.147 | 0.852 |
| Hybrid Nonsense | TBONTB | 0.310 | 0.797 |
| | New | 0.265 | 0.801 |

As in all cases the differences in metrics between studies are small, we accept them while acknowledging that the models
analyzed in this study are different than those in Acuña Espinoza et al. (2024). Moreover, the main objective of the present



study is not to set a "state-of-the-art" benchmark for a particular dataset and, accordingly, the overall message that we wish to communicate is not affected by the differences in performance between studies.

*Author contributions.* The original idea of the paper was developed by all authors. The codes were written by EAE and MAC. The simulations were conducted by MAC. Results were discussed by all authors. The draft was prepared by MAC and AG. Reviewing and editing was provided by all authors. Funding was acquired by AG and UE. All authors have read and agreed to the current version of the paper.

*Competing interests.* The authors have no competing interests to declare.

*Acknowledgements.* We acknowledge funding by the Deutsche Forschungsgemeinschaft (DFG, German Research Foundation) both under Germany's Excellence Strategy—EXC 2075–390740016 and the project 507884992. We would also like to thank Hoshin V. Gupta for his encouragement and thoughtful discussion of the results in this paper and their relationship to model complexity.



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



**Figure 2.** Didactic examples, demonstrating the evaluation of hybrid hydrological models by measuring the entropy of the model parameters and the LSTM hidden-state space. Left column: Schematic illustration of hybrid model structures, with Model 1 representing the "true" conceptual physically constrained model coupled with the LSTM as a reference. Center column: Segment of observed/predicted discharge time-series. Right column: Time-series of LSTM-predicted parameters and their univariate distributions.




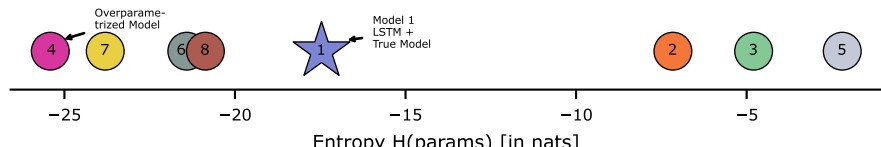

**Figure 3.** Benchmarking axis based on the entropy of the time-varying parameters in the different hybrid models (didactic examples).

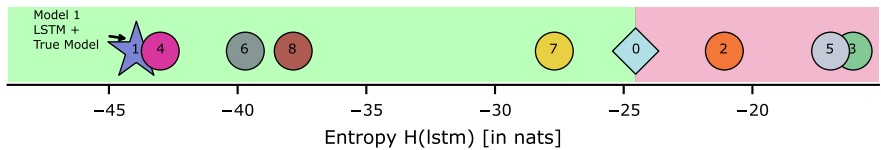

**Figure 4.** Benchmarking axis based on the entropy of the trajectories of the LSTM hidden states in the different hybrid models and the pure LSTM (didactic examples). The division of the green and red backgrounds serves to identify the addition of "good" and "bad" constraints, respectively.

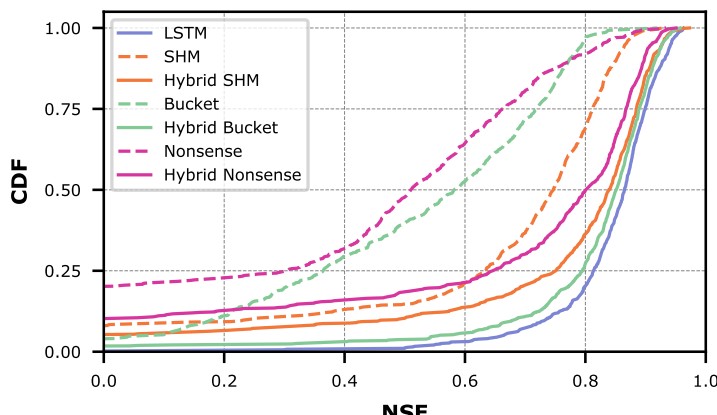

**Figure 5.** Comparison of model performance between conceptual models with static parameters (dashed lines), hybrid models with dynamic parameters (solid lines), and the pure LSTM for all CAMELS-GB basins.



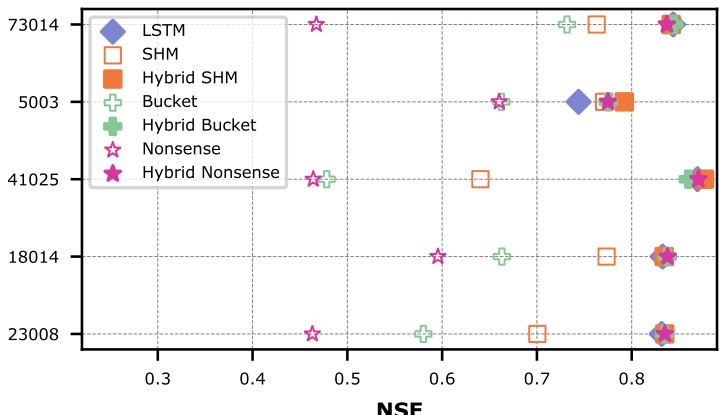

**Figure 6.** Comparison of model performance between conceptual models with static parameters, hybrid models with dynamic parameters, and the pure LSTM for individual CAMELS-GB basins.

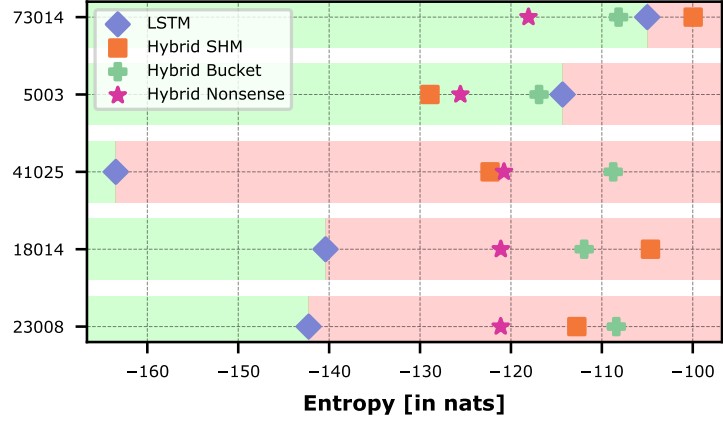

**Figure 7.** Entropy of the trajectories of the LSTM hidden states in the different hybrid models and the pure LSTM for individual CAMELS-GB basins. The division of green and red backgrounds matches that of Fig. 4.





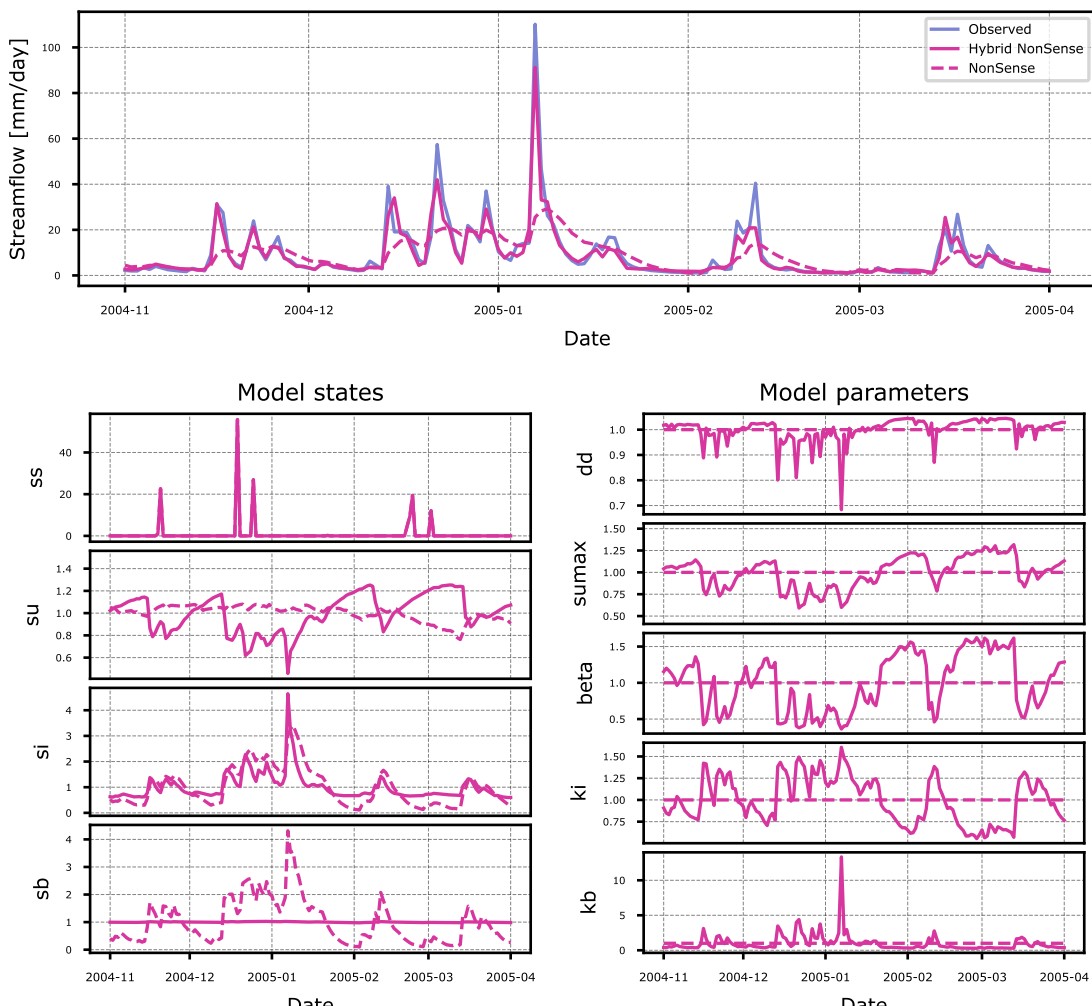

**Figure 8.** Differences between simulated streamflow, states and model parameters of the Nonsense and Hybrid Nonsense models for basin 73014. Both the states and model parameters are shown on a scale relative to their mean.



**Figure 9.** Differences between simulated streamflow, states and model parameters of the SHM and Hybrid SHM models for basin 73014. Both the states and model parameters are shown on a scale relative to their mean.




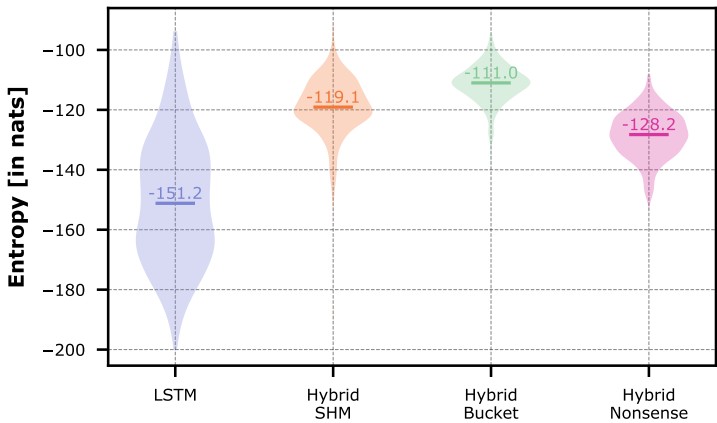

**Figure 10.** Violin plots of the entropy of the trajectories of the LSTM hidden states in the different hybrid models and the pure LSTM across all CAMELS-GB basins.

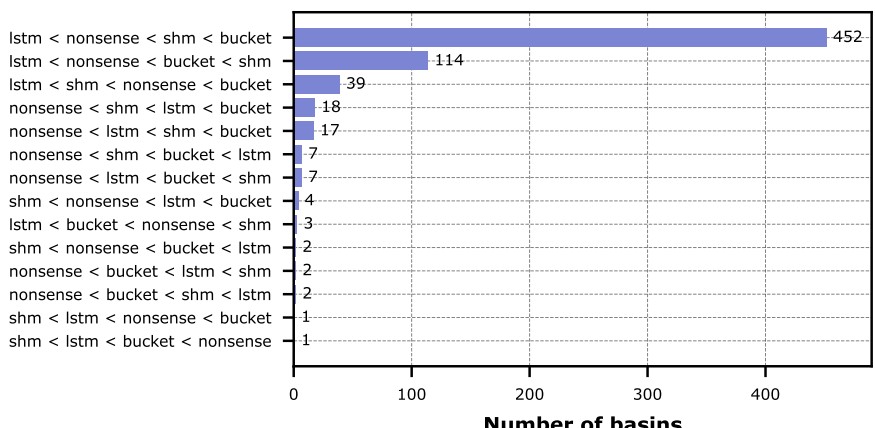

**Figure 11.** Counts of the different entropy-based model ranking outcomes across all CAMELS-GB basins. To limit the length of the label, the shorter conceptual names of the hybrid models were used but the counts are for the hybrid versions of these models.




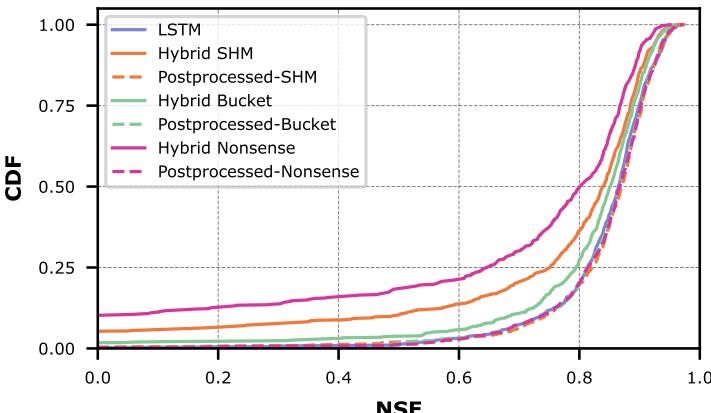

**Figure 12.** Comparison of model performance between conceptual models with static parameters (dashed lines), post-processing hybrid models with dynamic parameters (filled lines), and the pure LSTM for all CAMELS-GB basins.

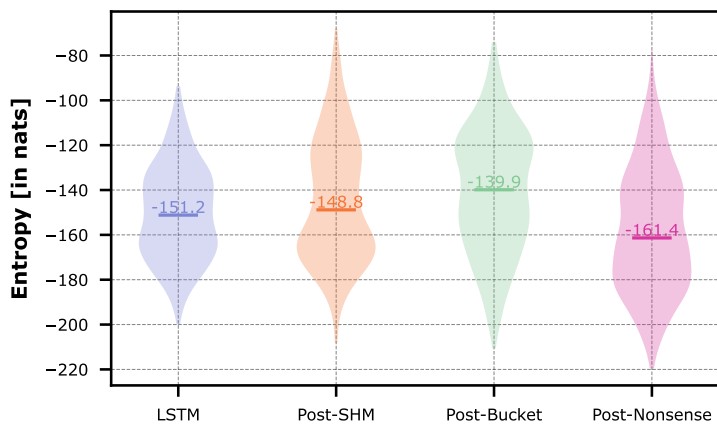

**Figure 13.** Violin plots of the entropy of the trajectories of the LSTM hidden states in the different post-processing hybrid models and the pure LSTM across all CAMELS-GB basins.



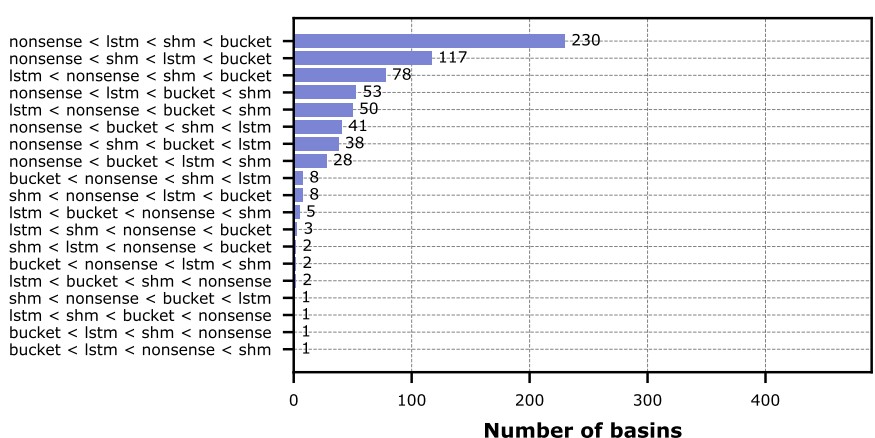

**Figure 14.** Counts of the different entropy-based model ranking outcomes across all CAMELS-GB basins. To limit the length of the label, the shorter conceptual names of the post-processing models were used but the counts are for the hybrid versions of these models.





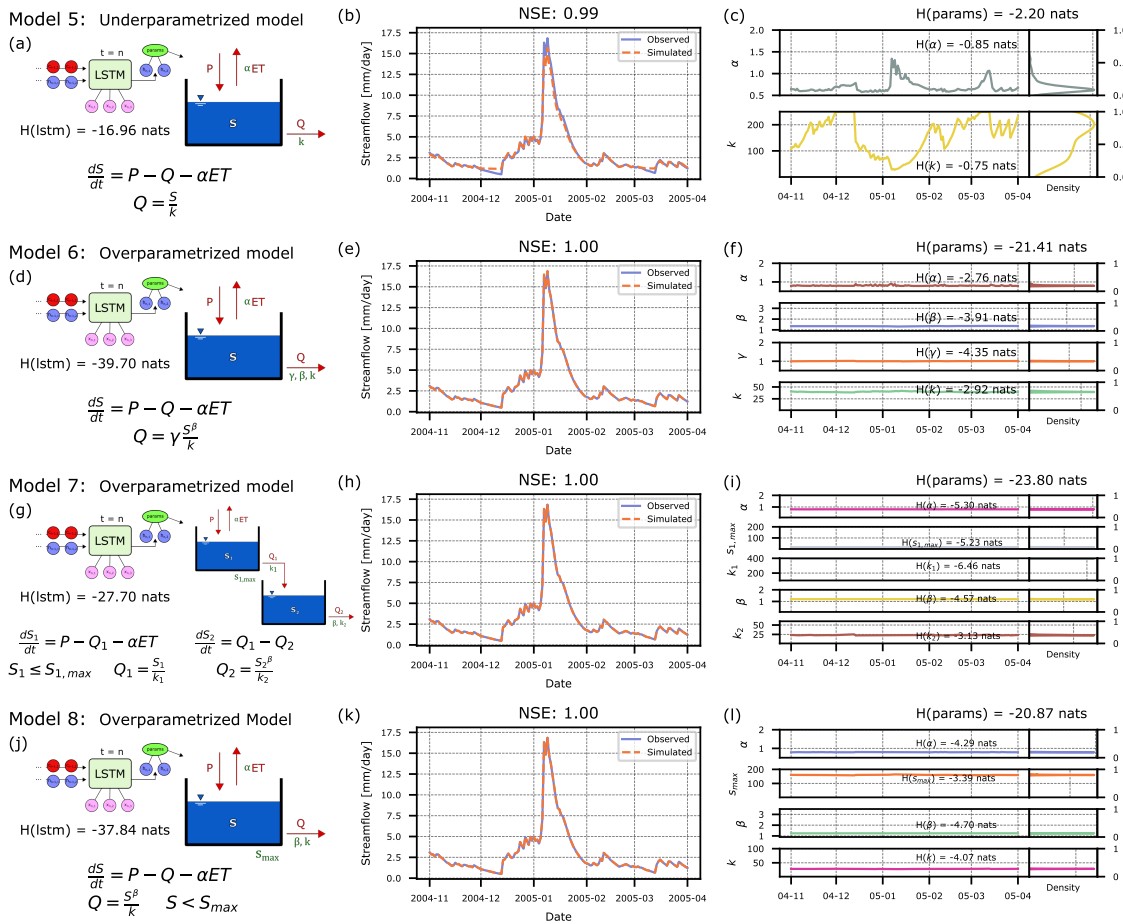

**Figure B1.** Additional examples on evaluating hybrid hydrological models by measuring the entropy of different model components



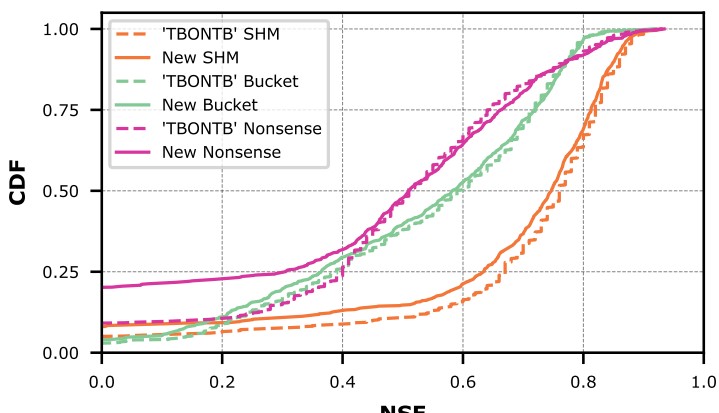

**Figure C1.** Comparison of model performance for models with static parameters between studies

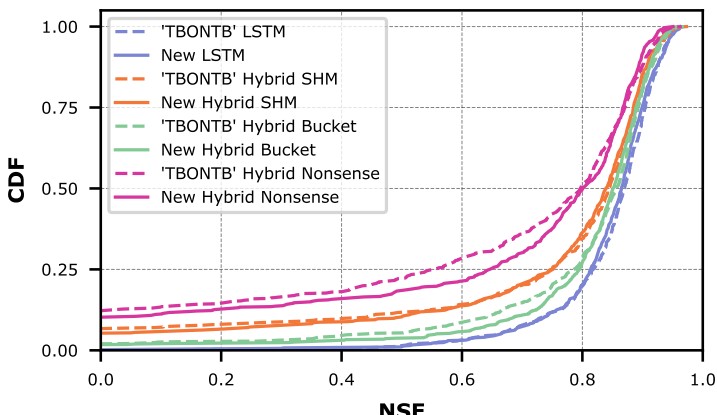

**Figure C2.** Comparison of model performance for models with dynamic parameters between studies