# Peer review of "When physics gets in the way: an entropy-based evaluation of conceptual constraints in hybrid hydrological models"

_EGUsphere, 2025_

## Author Comment (AC2)

**Reply to RC2 'Comment on egusphere-2025-1699', Anonymous Referee #2**

Review for HESS manuscript: When physics gets in the way: an entropy-based evaluation of conceptual constraints in hybrid hydrological models by Manuel Alvarez Chaves et al.

The topic and method discussed and used in this manuscript are interesting and timely—when and how the addition of physical principles genuinely enhances model performance and improves the representation of underlying physical processes. Overall, I am sensing some logical issues, making the findings might not be conclusive or fully compelling. They designed a synthetic case and a real application on CAMELS-GB dataset. In the synthetic case, they found that the LSTM is the dominant factor in the hybrid model and can even effectively adjust a nonsense formulation to perform as well as the other models. The LSTM (for parameter estimation in the hybrid model) applied for more accurate process-based model structure has less entropy. The nonsense model has the highest entropy. In the CAMELS-GB application, LSTM has the lowest entropy and highest performance, followed by the nonsense model structure with the second lowest entropy and the hybrid bucket with the second highest performance. With that, they made the conclusion that the connected physical model did not simplify the prediction nor improve performance.

We sincerely would like to thank the reviewer for their evaluation of our work and the time they have taken to engage with our manuscript, and we appreciate the interest in advancing hybrid modeling.

First we would like to clarify some points particularly for the synthetic case study as described by the reviewer. In the synthetic case study, we did not propose an architecture that was "nonsense" but rather conducted a controlled experiment where we generated synthetic data using a known "true" conceptual model, then tested eight different hybrid models as working hypotheses to see how well they captured the underlying data-generating process. When evaluated solely on their ability to fit the data, all eight hypotheses performed equally well in terms of MSE. However, we then applied our proposed entropy measurement to assess whether performance was achieved through the conceptual model component or through compensation by the LSTM component modifying the conceptual model's parameters.

Our key finding was that the hybrid model containing the "true" conceptual model required the least assistance from the LSTM (lowest entropy), while models with architectures very different from the true underlying process required the most assistance from the LSTM (highest entropy). This demonstrates that our entropy metric can distinguish between conceptual models that perform well for the right reasons versus those that achieve good performance just through compensation by a data-driven component, while traditional performance metrics cannot make this distinction.

Regarding the CAMELS-GB case study, you provide an accurate description of our entropy rankings, yet we want to emphasize a very important distinction: entropy is not equivalent to performance. While entropy measures how much the LSTM component must compensate for

the conceptual model, model selection should still consider traditional performance metrics like NSE. In our comparison across specific basins, we intentionally selected examples where all models achieved similar performance to match our synthetic case findings and focus on the insights which entropy can reveal.

It is not our intention to make definitive statements about overall model choice, and we will carefully go through a revised version of the manuscript to avoid this impression. The primary focus of this paper is on a novel model evaluation and comparison method that adds insights to better understand why specific hybrid versions perform well or not so well, rather than guiding a process of model selection. We will highlight this in our revision to make sure that our scope and conclusions are presented more clearly. We also go into more detailed specifics by addressing each of the points made in this review.

Here are my major thoughts on hybrid models, the experiments designed in this work, and the conclusions drawn from it

1. The LSTM in the hybrid model is used for parameter estimation, while the simulation is conducted by the process-based model (PBM). The main reason for incorporating PBMs is to improve model interpretability and performance, especially when data is limited (by reducing the searching space). The latter is achieved through physical constraints provided by PBMs. The physical constraints include not only mass balance for each bucket but also the interactions between fluxes. More specifically, the hybrid model provides clearer expressions (better interpretability) of the physical processes than black-box models (e.g., LSTM) and enables the calculation of internal variables. This means that we can incorporate more data as target values to further constrain or evaluate the model.

Many studies, including those by the authors, showed that the internal variables such as soil moisture and ET are comparable to observations. If the physics are totally incorrect (or, as the authors commented --- "physics-ignored"), how is it possible that these internal variables make sense? If some of the modules carry realism to some extent (which would make "physics-ignored" incorrect), can the authors' method ascertain what is good or correct? If the method cannot ascertain what is good, how can it separate the good from the bad?

Thank you for this comment. We would like to clarify that we use "ignored" rather than "incorrect" physics because this better describes what may happen in hybrid models. Physical laws such as conservation of mass and physical principles like the interaction of fluxes are indeed encoded in the model, but in some cases they might not be satisfied in the ways in which the modeller might expect.

If the prescribed constraints are not respected in the hybrid model, then also interpretability is not given as such. This emphasizes the need for and usefulness of our proposed method: we need to better understand if the hybrid model under investigation truly respects the physics as intended (in that case, interpretation and consistency tests make sense and are meaningful), or if the data-driven component effectively changes the structure - up to essentially manipulating all

processes, leaving only mass balance as effective physics constraint (in that case, interpretation based on the initially prescribed physical processes would be misleading and not suitable for advancing science).

We suspect that hybrid models, more often than not, show an effective structure that is notably different from the initially prescribed physics constraint. We hope to motivate the readership and community to perform extensive tests with our proposed methodology to confirm or dismiss this hypothesis.

The reason why so far this issue has not been widely documented might lie in the fact that even hybrid models that are constrained by "inadequate physics" have the capacity to show high correlations with unobserved variables. Our Hybrid Nonsense model demonstrates this clearly as was shown in Acuña Espinoza et al. (2024). Despite deliberately encoding nonsensical interactions in between its fluxes and storages, this model still achieved a correlation with soil moisture (from ERA5-Land data) of 0.85 that is nearly identical to the more physically reasonable Hybrid SHM with a correlation of 0.86. This shows that compensation by a neural network can make even poorly encoded knowledge of physics appear to work well. One possible explanation is that soil moisture dynamics are mostly driven by boundary conditions (rainfall - soil moisture increases, no rainfall - soil moisture decreases), making it not a particularly challenging test. It remains an open question how consistency checks should be designed to be truly informative in a physics context; our proposed method will help establish the ground for interpretation of such consistency checks by warning the modeler if the prescribed constraints are actively compensated for.

The key contribution of our work is hence providing a systematic way to distinguish when physics are consistently utilized versus ignored and manipulated. Our entropy metric, benchmarked against a purely data-driven baseline, enables this separation. Higher entropy than the baseline indicates "bad" physics, i.e. constraints that need compensation, while lower entropy indicates "good" physics, i.e. contributing meaningfully to model performance. "Good" and "bad" is used as abbreviation for these longer definitions in the manuscript and properly explained as such; if that is perceived as too bold, however, we would consider replacing those terms in the caption of Fig. 4.

Currently, the model in this study is trained only with streamflow data and evaluated against streamflow. However, the authors could design a more complex synthetic case with additional internal variables such as ET, snow, soil moisture, and baseflow, and explore the following:

   a. If we add more constraints to the internal variables (by using them as additional targets), can the nonsense formulation still achieve the same performance as the correct model?
   b. Can a model with a nonsense formulation provide accurate simulations of these internal variables when trained only on streamflow?
      In real cases, even when we lack direct observations for some variables, we can still compare model outputs to satellite or reanalysis data to gain insights into the accuracy of the internal variables and model structure.

c. When your hybrid models produce accurate predictions for both internal variables and streamflow, do you still observe an increase in entropy when adding physical models? Or, if entropy increases, can you still maintain the same level of accuracy?

Thank you for these suggestions. These are indeed questions we have considered ourselves and would be valuable to explore in future work. However, we believe it's important to maintain a focused scope for this study to ensure we provide a thorough answer to our central research question. The performance of hybrid models as such, and the best training strategy (single target vs. multi-objective) to increase it, are not of core interest here (as we noted above, the entropy metric is not to be confused with a performance score).

Furthermore, we do not wish this study and publication to become a comprehensive evaluation of the Nonsense model. The motivation for this study arose from the observation in Acuña Espinoza et al. (2024) where good predictions of streamflow were achieved without the need to add a proper physics-based model like SHM, but using Nonsense instead. This led us to investigate why you could seemingly add any conceptual model structure, even deliberately "nonsensical" ones, and still achieve equally good hybrid predictions of streamflow. We believe our current scope allows us to provide a solid understanding of this phenomenon through both our synthetic case study and the CAMELS-GB case study, and we see the questions you've raised as natural and important extensions for future research.

Regarding the broader question of using correlations with internal variables to validate hybrid models: as discussed above, there is a need to increase the rigor of such analyses by putting the correlation results into the proper context. While important hybrid modeling studies have used correlation of internal variables as validation (Feng et al., 2022), from the other perspective, it has been shown that LSTM cell states can also achieve strong correlations with observed variables, even with individual cells presumably learning to track specific processes like snow accumulation (Kratzert et al., 2019).

Instead, we first need to understand what the hybrid model is doing, before we can interpret correlations. It might be that the internal functioning of the hybrid model is substantially different from what the modeler had put in, and hence a surprisingly high correlation with variables not used for training might just mean that the hybrid model found its way into an effective time-lagged representation of how rainfall increases the level of a storage, before producing discharge: this is what we observed in our test case with the Nonsense model. We therefore see our proposed analysis as a first and necessary step to put any correlation analysis on solid grounds. The result of a correlation analysis, however, will not have an impact on our method in any sense, which is why we don't see merit in extending the scope of the current manuscript to internal variables. Also, the question how we can better enforce physics constraints to prevent them from being overwritten is beyond the scope of this study; rather, the proposed method provides a tool for modelers to test the effectiveness of their way of constraining hybrid models.

2. Should we make all parameters dynamic? Similar to traditional conceptual models, hybrid models also suffer from overfitting when too many parameters are calibrated (or "learned," in the

case of hybrid models), especially when done dynamically. We should not make all parameters dynamic. This caution has been raised in several recent works on hybrid modeling. By making all parameters dynamic, we give the LSTM too much freedom and intentionally allow it to dominate the hybrid model. One key point I want to emphasize is that LSTMs (and neural networks in general) are "lazy" models—they tend to find the most convenient (easiest) way to make predictions. Rather than learning the true functional relationships of parameters, they often memorize patterns in the time series, and often special effort is needed to make it extract the necessary information. To discourage this "laziness" in neural networks, two approaches can be considered:

    a. In addition to the temporal test as done in this work, prediction in ungauged basins or regions is a useful evaluation method. Verifying the accuracy of internal variables is another effective strategy. If a hybrid model demonstrates good spatial generalizability, how does its entropy compare to that of a pure LSTM?

    b. If we only allow a limited number of parameters to be dynamic, for example, just beta (shape coefficient in the soil moisture zone), can all models still achieve the same level of performance? How does their entropy compare to that of the pure LSTM?

The authors touch on a similar point in Lines 645–648: "Notably, this process occurs entirely under the hood. If we had evaluated performance using only NSE, we might have mistakenly concluded that the Nonsense constraint was just as valid as SHM or Bucket, since all three achieved the same performance when paired with the LSTM." I hope the authors can consider my suggestions in their entropy analysis experiments. The current experiments and narrative might unintentionally lead readers to conclude that the hybrid model lacks of effectiveness and interpretability.

Thank you for these suggestions. In fact, ultimately we wish to help improve the effectiveness and interpretability of hybrid models, by providing a rigorous analysis tool that helps looking under the hood. We challenge the assumption that hybrid models become effective simply because we add a specific structure like SHM. As reflected in the lines you highlighted, our intention is to demonstrate that, when predicting streamflow alone, some models might be good for the wrong reasons, and we aim to provide tools to identify such cases, to have a chance to correct them and develop better ways of building hybrid models.

We agree that not all parameters need to be dynamic, and our implementation actually provides the flexibility to keep certain parameters static while making others dynamic. Your suggestions represent excellent directions for future research. For example, if we freeze certain parameters, does the LSTM increase its reliance on remaining parameters (increasing entropy), or does it find more elegant solutions with the remaining degrees of freedom (reducing entropy)? These questions exemplify how this study opens new research avenues while advancing model evaluation beyond simple predictive performance to include model complexity, a topic that is typically not addressed.

Nevertheless, for this study we deliberately followed that of Acuña Espinoza et al. (2024), as we specifically wanted to address the questions those authors raised. To emphasize, we didn't do any new modeling for this study but analyzed previously published results.

With respect to prediction in ungauged basins: we focused on temporal rather than spatial extrapolation because we also followed the scope of Lees et al. (2021), which matches the standard benchmarking practice for CAMELS datasets. Theory-wise, we do not see the direct benefit of applying our entropy metric to the predictions of hybrid models in ungauged basins. The potential overwriting of physics constraints happens during the training phase, and hence it is natural and logical to analyze it in the respective basins that these data are available for. In a secondary step, one could of course also use our method to analyze the variability of LSTM hidden states when predicting in ungauged basins, but the interpretation becomes more difficult - we see this as an open question for future work.

Regarding your point about neural networks being "lazy" models, please indulge us as we'd like to offer a philosophical perspective. We believe this apparent "laziness" sometimes aligns beautifully with the principle of Occam's razor in that what appears to be lazy may actually represent profound insight that recognizes that a complex problem can have a surprisingly simple solution. In our case, the LSTM's ability to ignore unnecessary complexity in models like SHM and focus on what's truly essential for streamflow prediction could be seen as elegant, efficient and parsimonious problem-solving rather than laziness. The fact that the LSTM could transform even our Nonsense model into an effective predictor suggests it identified the core requirements for this prediction task, potentially revealing important insights about what's actually necessary for predicting streamflow accurately in this dataset. Further, we designed the didactic synthetic examples deliberately such that they could reveal if LSTMs tend to do "overcomplicated" things or fail by only memorizing time patterns. It was reassuring to see that they identified the simplest and truest representation possible.

3. I also felt the entropy argument has some logical issues: The fact that LSTM has the lowest entropy simply means it has no reason to care about processes. Let's think of process-based Earth System models with land surface processes like energy balance and carbon cycles, with many complicated calculations and quite likely lower NSE values. Now, let's say we replace a component with large variability but minor influence on streamflow with an NN-based dynamic parameter. The dynamic parameter will learn some of the missing dynamics but will generate a large entropy with a quite small gain on streamflow performance, but does this mean it is worse than a model that completely ignores these dynamics? Moving one step further, if one "downstream component" to the NN has large structural error and generates lots of noise, the NN ends up ignoring this component (giving it nearly 0 weight), and route information through other paths. The total entropy may end up being lower. Does this mean the second one is a more realistic model?

Thank you for raising these questions.

To reiterate, our measure of entropy is designed to evaluate whether added model components contribute meaningfully to make the task of prediction easier or whether the neural network must compensate for their inadequacy. Entropy is used as a diagnostic tool. When we argue that entropy relates to the challenge of prediction (lines 394-401), we're focusing on this relationship between model complexity and difficulty in prediction. Further, as stated above, we do not wish to provide a model selection method that pits performance against entropy. Rather, we provide a tool that transparently shows how much the LSTM has to compensate, which is expected to be very valuable in model evaluation. We do not see the point of providing a universal recipe of how to balance this with performance. The main driver of advances is, from our perspective, understanding why our models perform as they do.

So coming back to your specific example, we believe that if a hybrid model learns to bypass structurally flawed components, this provides valuable diagnostic information. While this doesn't make the model more "realistic" in terms of process representation, it reveals which processes actually contribute to predictive skill versus which introduce unhelpful complexity. Understanding when and why processes are ignored can guide us toward better representations or help recognize when we're asking models to represent processes they cannot adequately constrain with available data.

4. One should not think that the NNs in a hybrid model learn only true physics --- it is a mixture of true missing (to be learned) processes and compensation for structural, parametric and even numerical deficiencies. The baked-in assumptions serve as constraints to limit the searchable subspace, hopefully making the framework more generalizable and giving meaning to intermediate variables. The hope is that the signal overwhelms the noise or that we can extract useful insights from learning. We verify the meaning using additional observations.

My point is that one should base such analysis on a case-by-case basis and the conclusion obtained by the experimental design in this study may not generalize to another case. Entropy measures variability, not correctness. A model may need high variability in parameters due to data noise, poor observability, or inherent system variability—not necessarily because the constraints are wrong. The authors acknowledge uncertainty later (Sect. 4.3.2 and 4.5) but still lean heavily on this deterministic interpretation throughout. Equating low entropy with model adequacy and high entropy with "physics being ignored" oversimplifies complex interactions between model structure, data, and learning dynamics.

Thank you for this comment. In rainfall-runoff modeling, where we work with significant abstractions of real-world dynamics, we acknowledge that what we learn are not "true physics" but rather macro-scale insights about system behavior, no matter if we talk about conceptual or hybrid models.  Further, here the NN is not used to learn "true" physics, as these are assumed to be encoded by the conceptual model which serves as the head layer of the LSTM. Rather, the LSTM learns how to bias-correct the conceptual constraint to arrive at close-to-true physics.

As the Reviewer states, models will struggle with data noise, poor observability, or unresolved system variability, and this will have an impact on how much effort the LSTM has to put in to

make the rigid conceptual constraint perform well under these conditions. Hence, while it will be difficult to interpret individual entropy values as "high" or "low", the comparison with the reference entropy value of the pure LSTM, and potentially other hybrid model structures as done in our study, for a given catchment is highly informative. This is why we promote constructing at least parts of the model evaluation axis in Fig. 4, and why we look at basin-specific rankings in Fig. 11 to complement the results from simply comparing entropy values across basins in Fig. 10. It will be interesting indeed to further refine our understanding how "adversities" in the data of the basin to be modeled impact the difficulty of the modeling task and hence the entropy metric in future work.

We agree that evaluation should be case-by-case, which is why we promote a widely applicable method, not its analysis outcomes. We made a careful effort to both provide an overview of results for that specific large-sample data set, and additionally investigated specific basins in-depth for our analysis in Sections 4.2, 4.3.1, and 4.3.2. The purpose of these analyses is to let the readers gain intuition about the possible outcomes of our analysis method and their interpretation, such that they can apply the method to their cases of interest. It will be valuable to find out if our results generalize across many different large-sample datasets, and how they depend on the specific hybrid model architecture. Again, with this study, we hope to stimulate further research along these lines, to advance hybrid modeling in hydrology but as Reviewer 1 pointed out, as well in many other disciplines.

5. Ignoring causality. None of the analyzed information metrics measure causality, while causality is a key value brought in with the incorporation of physics. The physical component can ensure that higher temperature can lead to higher evaporation, or ensuring that precipitation (not humidity) drives runoff. The causality and baked-in sensitivity allow future climate projections to be more reliable. Your information metrics do not reflect such guarantees.

Thank you for this observation. You are correct that our entropy-based analysis does not measure causality, because it is not designed to do so, and we also do not claim anywhere in the text that it would. We do agree that what the Reviewer describes as causality is a desirable property and insightful methods should be developed to measure/detect it in hybrid models. This is an open question, yet unrelated to the topic and goal of our study.

Minor comments:

Line 102–104: There are several other studies that have compared hybrid models with LSTM. These could also be mentioned here for completeness.

Thank you for this suggestion. Indeed there are, and we will include some additional references to comparisons between hybrid models and LSTMs in Section 1.3 of a revised version of the manuscript, while keeping these lines focused on motivating the current study.

Line 451–453: This depends on which aspect of the problem you're examining. Hybrid models should be compared not only to LSTM but also to traditional process-based models. Given the same structure of a process-based model, the hybrid version can yield better performance. Compared to LSTM, hybrid models offer better interpretability. The fact that a hybrid model can make a nonsensical model perform well does not mean that a hybrid model built with a correct process-based structure is invalid. As I mentioned in my major comments, the model should be evaluated more comprehensively. Previous experience and domain expertise can help reduce equifinality problems and prevent obviously nonsensical model configurations.

Thank you for this comment. We agree that hybrid models may provide valuable capabilities between traditional and data-driven approaches, and we are not saying that well-designed hybrid models are invalid. As we have pointed out before, it is critical to understand if the prescribed physics are obeyed or not, before making a claim about interpretability.. The fact that our "nonsensical" structure could be transformed into an effective predictor suggests we should pay more attention to what is effectively happening in the hybrid model structure, and be more critical about whether what we're adding actually contributes to the task at hand. Our entropy analysis is intended as a complementary tool to help identify when neural networks are compensating for structural choices, allowing more informed decisions about the merits (such as interpretability) of a model.

Line 561–562: This supports my view that LSTM is a "lazy" model. It tends to find ways to bypass physical constraints, especially when dynamic parameters are allowed.

While we wouldn't necessarily agree with the notion of "lazy" (we rather find it smart and reassuring that the LSTM finds a parsimonious, efficient and skillful representation of the system), we fully agree that the LSTM finds ways to bypass physical constraints if they are not helpful for the prediction task, and this is exactly what we shed light on with our method - in a quantitative way by means of our entropy metric, and qualitatively by analyzing the overwriting behavior of the dynamic parameters.

Line 637–638: A simplified task means that LSTM has found an easier path to make predictions, but this does not necessarily indicate higher accuracy. Multiple evaluation metrics should be used to support conclusions

We disagree with this point. All models were trained and evaluated using the same loss function (NSE or MSE), ensuring accuracy is directly comparable. So the LSTM indeed achieved higher accuracy in streamflow prediction. What we refer to as a "simplified task" is prediction with lower entropy: if the physics constraint took over main parts of prediction, the hidden states in the LSTM wouldn't have to vary as much as in the pure LSTM, and hence entropy would be lower. While additional metrics could provide more detail and further insight, our study focuses specifically on the "effort" of the LSTM in this hybrid model architecture.

**References**

- Acuña Espinoza, E., Loritz, R., Álvarez Chaves, M., Bäuerle, N., & Ehret, U. (2024). To bucket or not to bucket? Analyzing the performance and interpretability of hybrid hydrological models with dynamic parameterization. *Hydrology and Earth System Sciences, 28*(12), 2705–2719. https://doi.org/10.5194/hess-28-2705-2024
- Feng, D., Liu, J., Lawson, K., & Shen, C. (2022). Differentiable, Learnable, Regionalized Process‐Based Models With Multiphysical Outputs can Approach State‐Of‐The‐Art Hydrologic Prediction Accuracy. *Water Resources Research, 58*(10), e2022WR032404. https://doi.org/10.1029/2022WR032404
- Kratzert, F., Herrnegger, M., Klotz, D., Hochreiter, S., & Klambauer, G. (2019). NeuralHydrology—Interpreting LSTMs in Hydrology. *arXiv:1903.07903* [Physics, Stat], 11700, 347–362. https://doi.org/10.1007/978-3-030-28954-6_19
- Lees, T., Buechel, M., Anderson, B., Slater, L., Reece, S., Coxon, G., & Dadson, S. J. (2021). Benchmarking data-driven rainfall–runoff models in Great Britain: A comparison of long short-term memory (LSTM)-based models with four lumped conceptual models. *Hydrology and Earth System Sciences, 25*(10), 5517–5534. https://doi.org/10.5194/hess-25-5517-2021

---

## Author Response (AR1)

**Author's response**

Dear Dr. Fenicia,

Thank you for handling our manuscript and enabling our discussion with the reviewers.

Alongside this response letter, please find a new version of our manuscript which incorporates the feedback received during the review process as well as a version of the manuscript with tracked changes. This response letter is organized in two sections. First, we provide a brief summary of the key points made by both reviewers and how they are addressed in this new version. Then, we include point-by-point responses to each review, recalling our direct replies during the discussion phase in light grey and indicating relevant changes made in the manuscript in black font.

**Scope**

We would first like to clarify the scope of this study because both RC1 and RC2 suggested additional experiments which would require modifying the setup in our case study. However, our experimental setup is intentionally constrained because we're building directly on "To bucket or not to bucket" (Acuña Espinoza, et al. 2024) where the authors raise the question of why a deliberately "nonsense" model was able to accurately predict streamflow. We answer this question by introducing a new metric based on entropy. This metric is able to distinguish if the predictive performance of a model (in terms of the objective function it was calibrated to optimize) comes from its data-driven or its physics-based component. We show what entropy can do in a controlled experiment and in a real case study while acknowledging the limitations that it has in the latter. We also advocate for data-driven models to be used as baselines from which a researcher can learn if modifications they are making or if the prior knowledge they are adding actually helps.

**Introduction and Context**

Both RC1 and RC2 requested for the introduction to be expanded. RC1 focused on providing more context on the evolution of hybrid models and the pros and cons of each modeling type (conceptual, data-driven and hybrid models). RC2 requested for more references that compare specifically hybrid models and LSTMs. To address these points, we have made changes accordingly to Sections 1.2 and 1.3 in the revised manuscript.

**Conclusions and Implications**

The main concern of RC2 is that the current experiments and narrative might unintentionally lead readers to conclude that hybrid models lack effectiveness and interpretability.
While we acknowledge that point and have carefully checked/revised our phrasing in several sections (mainly Sect. 4.3.3 and the Conclusions), we would like to emphasize that our intention is actually to help improve effectiveness and interpretability by providing a rigorous analysis tool that reveals potential pitfalls that lead to non-interpretability, but under the hood. Our

contribution allows users to identify when models are being right for the wrong reasons, and we provide tools to help correct them.

**Focus on Streamflow**

Both RC1 and RC2 noted that the study focuses on streamflow prediction. RC1's perspective was that by focusing on NSE and entropy, this potentially misrepresents broader hybrid modeling goals such as interpretability, process fidelity and multi-variable outputs. While it is true that we didn't consider multi-variable training, in Sect. 4.3.2 we do focus heavily on the idea that one should verify that an imposed constraint is still valid to fulfill the desire of interpretability and process fidelity. This is highlighted in several other places of the manuscript. Additionally, RC1 indicated that focusing on streamflow may reinforce issues of equifinality. While this is explored in the synthetic cases by adequately fitting the parameters of our "true" model, we don't acknowledge it explicitly because we're exploring this connection in a separate study. We believe that such theoretical and methodological implications extend beyond the scope of the current manuscript.

RC2 suggested more synthetic cases to evaluate if a "nonsense" formulation still performs well, but that is not the intention of the synthetic cases. The objective of the synthetic cases is to reframe the experimental setup of Acuña Espinoza et al. (2024) in a controlled setting and translate our findings to the results of the real case study.

We sincerely thank you and the reviewers again for the valuable comments. We believe that the modifications we made to the manuscript and our previous direct replies to their feedback, effectively address the reviewers' comments, concerns and suggestions.

Kind regards,
Manuel Álvarez Chaves on behalf of the co-authors

**Review by Georgios Blougouras & Shijie Jiang (co-review team)**

General comment 1: According to the phrasing of the paper in many instances, the readers might be led to believe that hybrid modeling is used here as a way to help the LSTM predictive performance (e.g., L106-107: 'While purely data-driven… genuinely enhances model performance', or, L621: '...reduce the effort required…'). However, the LSTM's role in the hybrid architectures explored in the manuscript is not to 'lead the predictions', but rather to infer the conceptual model parameters. Therefore, describing the models as 'physics-constrained' (L9) might not be the most accurate description - maybe something like ML-enhanced/parameter learning/… differentiable modeling would be more fitting. The change in terminology (although potentially annoying), implies different perspectives regarding the role of the data-driven and the physics-driven components of the model. To be more precise, the manuscript explores if different architectures are 'helping the LSTM', even though to begin with, the concept of parameter learning usually reflects the opposite direction (i.e., exploiting the LSTM to help the conceptual model improve its predictions). To my opinion, this creates a 'conceptual mismatch' across the perspective of the manuscript and the 'typical' way such hybrid models are used. I encourage the authors to clarify if their framing is applicable or not under this context - maybe it might be more accurate to view the LSTM as being 'constrained', rather than 'assisted' by the conceptual model. In light of this architecture interpretation, it would be helpful for the authors to revisit their interpretation of the entropy-based results, to ensure that the evaluation and subsequent conclusions are aligned with the actual model structure and what it represents.

Thank you for this thoughtful comment about terminology and framing.

Indeed, we show two different modeling setups: Section 4.3 shows the results of parameter learning to enhance a conceptual model as the reviewer describes it, and Section 4.4 shows the results for post-processing a regular conceptual model to improve performance of a data-driven model. As these are two very different approaches and there exists no rigorous definition of what a "hybrid model" is, we see this name as an apt description of what we did in both cases and we would like to keep it as such.

Both perspectives, led by the data-driven component and physics-constrained, are valid and depend on the researcher's background and starting point. A physics-based modeler would naturally view our Hybrid SHM as a conceptual model with dynamic parameters, while a data-driven modeler would see it as a constrained LSTM. Further, our analysis results indeed suggests rethinking the terminology: if one starts from the more traditional hydrological viewpoint of improving a conceptual hydrological model by letting an LSTM control its parameter values, the performance results with the pure LSTM scoring best in most cases will cause the modeler to frame the question rather from the LSTM side: is the additional conceptual part in any way helpful (i.e. constraining in the right way, or assisting), or rather making the prediction problem harder? In a revised version of the paper, we will carefully check the order in which we introduce and use the different terminology, and make sure that it is in proper context with our premises and results.

For the hybrid models which focus on parameter learning we do see the role of the LSTM as leading predictions. As an example: because Hybrid Nonsense modifies 'su' and 'si' to behave as time lags for the output, the outflow is mostly managed by the interaction between 'sb' and 'kb'. Considering the previous point, in Figure 8 we see that for the high flow peak that happens in 2005-01, 'kb' is increased disproportionately just so that the model can match the peak based on the volume available in 'sb'.

The role of the LSTM as leading predictions is more clear for the post-processing models, as the predictions are given directly by the data-based component being informed by the results of an initial run of the physics-based component. Therefore we do see the two setups as cases in which the conceptual model assists (or doesn't) the LSTM. Moreover, particularly the setup that uses parameter learning can be effectively described as physics-constrained because conservation of mass is imposed by the conceptual model at the end of the pipeline. Therefore we consider that both terms: 'constrained' and 'assisted', apply.

We added additional commentary on the 'sb' and 'kb' behavior in Figure 8 to better illustrate how the LSTM leads predictions, but we believe our current terminology accurately reflects the model architectures and their roles.

Our case study has two different modeling setups: Sect. 4.3 shows the results of parameter learning to enhance a conceptual model, and Sect. 4.4 shows the results of post-processing a regular conceptual model to improve performance of a data-driven model. This was clarified in Sect. 1.4, and there is additional discussion in Sect. 4.3.2 to highlight how the LSTM indeed leads predictions.

General comment 2: The manuscript defines the value of hybrid modeling primarily in terms of streamflow prediction, measuring the "utility" of physics constraints by the degree to which they reduce LSTM parameter variability (entropy) for this single task. In my view, this approach risks narrowing the broader motivation for hybrid models, which is not limited to improving runoff prediction or reducing model complexity, but also includes enabling more meaningful and diagnostically useful process representations (e.g., for ET, soil moisture, or system anomalies). I am concerned that by focusing only on this narrow predictive context and a single diagnostic (entropy), the study may misrepresent the broader role of physical constraints in hybrid models. Many would consider physical constraints valuable even when predictive skill does not improve, as they support interpretability and process fidelity. I recommend that the authors more explicitly clarify the intended scope of their analysis and discuss the limitations of generalizing these findings to other goals of hybrid modeling.

Our focus on streamflow prediction reflects what we observe as the predominant approach in current literature. As discussed in Sections 1.4 and 4.1, many existing hybrid modeling studies emphasize predictive performance for a single target, often without detailed analysis of why specific physical constraints or conceptual components are effective relative to alternatives.

Rather than dismissing the broader value of hybrid models for interpretability and process fidelity, our study aims to provide a methodological framework for systematically evaluating the degree to which physical constraints actually constrain model behavior. This is highly relevant also in the context of interpretability and process fidelity: if the intended model structure that shall guarantee interpretability is overwritten, this argument is no longer valid or we might actually find a better (in our case: simpler) process representation that again allows for interpretability and scientific learning. While we focus on entropy as our primary metric, we believe entropy complements, rather than replaces, other evaluation approaches focused on process representation and interpretability, as we demonstrate in Section 4.3.2. We appreciate the perspective expressed by the reviewer in this comment and will include these considerations as an introductory point for this section in a revised version of the manuscript.

Our focus on predicting streamflow follows from what we observe in current literature and our reference study. We extended Sect. 1.3 to incorporate the ideas of interpretability of hybrid models and correlations between model states and unobserved variables, and address these ideas in Sect. 1.4, Sect. 4.3.2 and Sect. 5.

General comment 3: In the current setup, the LSTM is only used to infer parameters, and streamflow is the sole observation timeseries used for evaluation. In my view, this setup may further amplify the well-known problem of equifinality in hydrological modeling. From this perspective, I am not convinced that low LSTM entropy necessarily reflects a physically meaningful or faithful conceptual model; rather, it may simply mean that the LSTM has found a convenient way to "satisfy" the runoff constraint, possibly by exploiting compensatory effects among parameters or model components. While the manuscript explores this issue by visualizing the distributions of the LSTM-predicted parameters (e.g., in Fig2 and Fig9), I would encourage the authors to discuss more explicitly how equifinality in particular might influence the interpretation of their results.

Thank you for this insightful comment about equifinality. We do explore this phenomenon to some degree in the synthetic case study which features overparameterized models, and there we found that the LSTM identifies rather robust and simpler solutions. Yet, generally, constraining a multi-parameter model by a single objective inevitably comes with the risk of equifinality. This issue is well-known in hydrology and it has been for a long time, therefore our study setup is consistent and comparable with existing practice. We agree that moving on to higher-variate constraints is desirable, yet we prefer not commenting on this in this manuscript because we are currently exploring this connection in a separate study, as the theoretical and methodological implications extend beyond the scope of the current manuscript.

While equifinality didn't present a problem in Sect. 3.2, we acknowledge the calibrating/training only on streamflow may reinforce this as an issue. Nevertheless, we prefer not to elaborate further on equifinality in this manuscript since we're currently exploring the relationship between it and entropy in a separate study.

**Specific Comments**

Introduction: Why are you mentioning the modular hydrological frameworks more than once (L31, L47,...)? To my understanding, there is no clear and direct follow-up regarding such efforts in the rest of the manuscript (though one could make a conceptual link, but it would still benefit from a clarification from the authors).

Appendix B: It would be helpful to add some more context/information about the design of the additional models.

If we see SHM as a proxy for a typical conceptual hydrological model, Bucket and Nonsense arise as alternatives that come from modular hydrological frameworks and the value of multiple working hypotheses even in a setting in which they are combined with neural networks.
In a new version of the manuscript, we will reference these ideas again in Section 2.2.2 where the hybrid models of the study are introduced. Also, inspired by this comment, we plan to place our findings in the broader context of modular hydrological frameworks in the conclusion (what can we learn from our results with respect to building conceptual models).

We believe that there is a clear motivation and explanation for each model, but in a revised version of the manuscript will acknowledge that they were inspired by the idea of having multiple working hypotheses and flexible modeling frameworks.

These ideas of modular frameworks are highlighted in Sect 2.2.2 because we have multiple models built on these principles in the synthetic and real case studies. We clarify this in Sect. 2.2.2 and Appendix B.

For the '1.3 Hybrid models' subsection: To someone unfamiliar with such modelling efforts, I think the current subsection lacks a bit of clarity - why should people care about hybrid models, and how have people employed them in the (recent) past? I think a little bit more context is required in this section (especially given that in the end, from your findings and conclusions, the paper would appeal not only to 'hybrid modellers', but also to the general hydrological modeling community). In general, every 'modeling introduction' subsection (i.e., 1.1, 1.2, 1.3) should have clear pros and cons of using each modeling type - this is not clear for section 1.3. Furthermore, additional context would be beneficial regarding the hydrological community's evolution from 1.1 to 1.2 and then 1.3. Why did LSTM become the 'benchmark' for many researchers in the last 5 years? Why do more and more hydrologist attempt to use hybrid models instead of pure data-driven models? Some of these questions are already answered in the text, but in a way that in my opinion might not be extremely clear to someone that has is not well familiarized with hybrid modeling.

We believe the points you've raised are addressed within the current text: motivations for hybrid models appear in lines 85-87 and 97-99, the rise of LSTMs as benchmarks is discussed in lines 56-60, and the shift toward hybrid approaches is covered in lines 71-74.

Moreover, this study builds directly on existing hybrid modeling literature and serves as a follow-up to Acuña Espinoza et al. (2024). We designed it for readers already familiar with hybrid modeling approaches rather than as an introductory text for newcomers. Given this context and scope, we believe the current level of background information is appropriate for our intended audience.

Yet, we acknowledge your point and, in line with our previous answers, in a new version of the manuscript we will broaden Section 1.3 to highlight additional benefits of hybrid modeling beyond performance improvements, which may help contextualize our work for a wider readership.

We believe some of these points are already addressed in the manuscript. Specifically about hybrid models, Sect. 1.3 was expanded based on this feedback.

L106: Do you mean 'data-driven components'? Because the hydrological models inherently are 'physics'-driven models. Otherwise, please revise.

L105-109: This whole text passage has the point of view of someone who wants to improve data-driven hydrological models by incorporating physical principles. At the same time, the opposite pathway (trying to improve physical models by using data-driven modules) is also common in hydrology, and can also benefit from your suggested contributions. Why not mention it here as well?

The statement should read: "They demonstrate that incorporating physics-based components or prior knowledge doesn't yield an improvement in model performance *over the data-driven solution*". We will modify it in a new version of the manuscript.

Thank you for the suggestion, indeed our proposed method can go both ways. We will add this statement in a new version of the manuscript.

These lines were rewritten to incorporate this feedback.

L110-111 (contribution 1): This quantitative metric is not 'self-standing', to my understanding, but relevant to a 'benchmark' model (in this case, the pure LSTM), right?. I believe it would benefit the clarity of the manuscript if the authors were explicit about this here

Indeed, the metric requires a benchmark data-driven model to serve as a baseline. We will add "in comparison to a purely data-driven benchmark" at the end of this sentence in a revised version of the manuscript.

Contribution 1 now includes "... in comparison to a purely data-driven benchmark".

L114-115 (contribution 3): This suggested contribution is somewhat unclear to understand under the provided context - a reader would need to read the following manuscript first to fully grasp what the authors mean by 'effective' and 'prescribed'. I suggest revising this.

Thank you for the suggestion. To clarify, we will add "based on the conceptual model" at the end of this sentence in a revised version of the manuscript. This is what we refer to as the prescribed structure.

Contribution 3 now includes "...based on the imposed".

L122: 'favoring' -> again, the context is missing at this point for the readers to fully understand what 'favoring' the data-driven component means, and I would suggest revising this a little bit to provide more information.

Thank you for the suggestion. We will change this sentence to "High entropy points to an imbalance in which the data-driven component compensates for inadequacies in the conceptual model by manipulating its parameters..." in a revised version of the manuscript.

The new statement reads "High entropy points to an imbalance where the data-driven component compensates for inadequacies in the conceptual model by manipulating its parameters".

L173: Similar to my point regarding the 1.3 section, I think the authors should provide a little bit more context about why this revival of dynamic parameters has happened in hydrological modeling. I would refer more to Tsai et al. here (https://doi.org/10.1038/s41467-021-26107-z), which is already cited in your paper regardless.

Thank you for the suggestion. Yes, based on this and the previous comment, we will extend Section 1.3.

Section 1.3 was expanded to incorporate this feedback.

L174 / Figure 1: To my understanding, the different model setups are not yet utilized, and they do not become fully relevant until section 3 (which could be viewed as a gray zone between a 'methods' and a 'results' section). It is fine if the model setups remain in Figure 1 and this section, but then the authors should also present what the individual models represent, because there are 5 subfigures in figure 1, leaving a lot of questions to the readers (especially the ones not familiar with past group efforts utilizing the SHM and Nonsense models). Otherwise, the authors could immediately move the figure and refer to it in a more relevant section.

Thank you for your suggestion. We will add a brief description of each model here as we believe this is a good place for Figure 1.

Descriptions for each model were added to Sect. 2.2.2 referencing Figure 1.

NSE* calculation: In the original work, this metric is cross-basin and there was an additional sum in the formula (over all basins, which does not exist here), while Kratzert et al. divided by the number of basins instead of the number of days. Here you use a basin-average metric,

deviating from the original formula as far as I can see (but please correct me if I am mistaken). Does this imply that the evaluation is done on a basin-scale? It would confuse/conflict with the rest of your paper where you indicate that you train on multiple basins.

Indeed there is a difference and we will clarify in a revised version of the manuscript.

In the way we wrote Equation 4, we are not including the terms related to batch averaging. In our training pipeline, NSE* is calculated per training batch where each batch contains training samples from a single basin letting us also calculate $s_i$ per batch without having to consider standard deviations from more than one basin. Training on multiple basins happens at the end of a training epoch where we average the loss function for all batches. This is different from how the expression is shown in Kratzert et al. (2019b) but both loss functions are functionally equivalent.

Equation 4 was modified to include batch averaging.

Section 2.3: I am concerned that the entropy metric may depend strongly on the specific architecture and hyperparameters of the LSTM. Have you tested how robust the entropy-based findings are to changes in LSTM design (e.g., number of hidden units) or to using alternative machine learning models? I am curious to what extent the conclusions hold beyond the particular ML setup chosen in this study.

It is correct that the absolute values of entropy depend on the chosen ML architecture; however, the qualitative ranking between the different hybrid models remains stable. We confirmed this by previous analyses where we initially applied our method to four models using five hidden states and obtained the same qualitative ranking. When we extended the analysis to eight models, we increased the hidden nodes to ten to achieve comparable performance across all models. This is crucial because, in practice, a researcher typically selects an LSTM architecture to optimize *performance* on their chosen loss function. Since our entropy analysis is applied post hoc to trained models, the key requirement is that all models use the same architecture, rather than the specific architectural choices themselves.

No changes were made to the manuscript based on this comment. The concern was addressed as a reply to the reviewer.

L200: Here I would suggest a small change in the phrasing - maybe emphasize that you move away from analyzing the entropy metric of the predicted parameter values, but nevertheless exploring (and visualizing) the LSTM predicted parameters is an important step in exploring the 'under-the-hood' performance of the model. Currently it reads as if you will completely ignore the information from the LSTM-predicted parameters.

Good point and thank you for your suggestion! We will change the phrasing in a revised version of the manuscript.

This sentence was removed and clarified in the following paragraph.

L214: Mention that the link for the UNITE toolbox is found in the 'code availability' section.

We embedded a link to the text, but you make a better suggestion. Thank you! We will change it in a revised version of the manuscript.

The statement now reads "... which can be found in the *code availability* section of this article".

L233 -> Appendix A1: please specify how these parameter ranges can be justified (you mention Beck et al., 2016, 2020 but later on in the text, leaving an open question for now).

There is actually a mistake in how the choice of parameters is described. The parameters from Beck et al. (2016, 2020) are for the case study in CAMELS-GB. From this reference, we adapted the ranges to fit the simpler models in the didactic examples. We will rectify this and clarify our choices in a revised version of the manuscript.

The reference to Beck et al. (2016, 2020) was moved to Appendix A2 mentioning that they are directly used in the CAMELS-GB case study and simplified for the synthetic case study.

Section 3.2: I think it would be beneficial to be more clear as to why you use a stand-alone LSTM model to compare against the hybrid setups (for example, L727-L730 of Appendix B could be mentioned here).

Thank you for your suggestion! We will move the description of Model 0 to the end of Section 3.1 as we're also advocating for an initial purely data-driven reference model.

Section 3.2 now includes a description of the results for Model 0.

L274: I would say that randomly selecting 5 basins and not repeating the experiment might not convince many large-sample hydrologists - especially if you do not elaborate on these random catchments or their representativeness… One could repeat the exact same experiment by using 5 other randomly selected basins (no need to go too much into detail, providing the results in the supplementary and the logs on the repository is more than enough), or further demonstrate why conducting an experiment on a set of 5 randomly selected catchments is more than enough to derive safe and transferable conclusions.

A key aspect of our didactic example is that we use synthetically generated streamflow time series from a conceptual model with known static parameters, rather than observed streamflow data from real basins. In this synthetic framework, the specific characteristics or representativeness of the selected basins are less critical because the "observed" streamflow is generated using controlled parameters rather than reflecting actual basin behavior. As such, we could have sourced the rainfall data from anywhere or even used a stochastic process to

generate it for this example. This is noted in L279 but we will stress this point in a revised version of the manuscript.

The reason for selecting five basins are now clarified in Sect. 3.2.

L362: I think mentioning results about models appearing in the supplementary (and especially figures that have additional results from models not yet revealed before the supplementary information) can lead to unnecessary complication. Maybe you can either remove them from the main figures and reveal them directly in the supplementary information. Alternatively, if you wish to keep them, maybe some additional information about models 6,7, and 8 in the main manuscript would be helpful. Furthermore, you could also have specific shapes for Fig. 3 and 4 representing each type of 'wrong model type' - this would be practical if you wish to keep models 6, 7 and 8 in the main figures (e.g., over-parameterized models have a 'square' shape or a certain color, 'wrong architecture' models have striped color, etc…). This is not necessary to be done, just a suggestion / visualization experiment.

We appreciate the suggestion, however we would like to keep it as it is right now. We provide a description of how these models resemble those in the main text in L361-363, and we decided not to provide a longer description because of their similarity to the models described in more detail previously in this section. The suggestion for the change in visualization is good, but there is an overlap as overparametrized models can also have wrong processes or architecture. These categories are merely descriptive and we don't think it's necessary to modify the axis.

No changes were made to the manuscript based on this comment.

L380: 'matches the true system' (and many other instances in section 3 where the word 'true' is used) -> I feel like the phrasing is a bit misleading, as true is currently an idealized setup. Of course you have mentioned this in the manuscript, but still the word choice is important, and someone could make the implicit connection that these models can well capture the real word 'true' signal, which is not the case as we see in section 4.

Thank you for the suggestion. The first instance of the use of the word *true* to describe the idealized setup in Section 3 happens in L235 and we use quotations to emphasize that this is a case of synthetic "truth". We reinforce this distinction in L498 for the real world case study. We believe this establishes context and a reader will keep this in mind while reading this section. While we appreciate the concern about terminology, as this issue is more philosophical, we prefer to maintain our current usage of "true" for the synthetic reference as it better serves the pedagogical aspect of this section.

No changes were made to the manuscript based on this comment.

L383: clarify which LSTM based entropy you refer to here - hidden-state or parameter based, because the distinction is important and it needs to be clear to the readers.

Thank you, we will add "hidden states" in a revised version of the manuscript.

The new statement reads "... LSTM *hidden states* …".

'On the Complexity of the Prediction Task' -> I think this can be seamlessly merged with the 3.4 subsection, or go directly into the appendix - I feel like it doesn't provide enough information as a stand-alone part of the manuscript…

We would prefer to keep this section as it reflects our interest in data-driven baselines and the analogy with complexity will appeal to some readers. Moreover, we highlight that our approach focuses on the data-driven component and measuring the entropy of the conceptual head layer is an open challenge that we hope can be addressed in the future.

No changes were made to the manuscript based on this comment.

Subsection 3.4 'Summary of the proposed approach' and overall point about Section 3: Now all the models had a (almost) perfect fit. This makes the evaluation 'easier', because now we know if the LSTM had to work 'overtime' to 'save' the model performance. But how can we evaluate this in cases where the models do not have equally comparable performance metrics? In other words, how do we measure the 'effort' by the LSTM to 'save' the streamflow prediction, if the final models do not predict streamflow prediction equally well? I am aware you touch this a little bit on the next chapter, but I think additional discussion on this matter on Subsection 3.4 would be very helpful - I am sure many readers would have this question.

We agree that this is a practical consideration that readers will share and, in a revised version of the manuscript, we will add additional commentary to point towards the relevant section in this subsection.

However we would like to emphasize that this is a didactic example which is predicated on having equal performance because we're not making any specific statements about model selection or preference. Specifically we are using this section to apply our proposed method in a controlled setting, and derive insights about the amount of effort prediction takes without considering any differences in performance.

In practice, we do consider that any kind of ultimate selection should be based not only on performance and entropy, but also on other criteria such as explainability, computational cost, etc. This is why we highlight very specific cases in Section 4.3.

The last paragraph of Sect. 3.4 was modified to incorporate this feedback.

L454-459: Here, you could mention that a description of these models (SHM, Bucket and Nonsense) can also be found later in this manuscript.

Thank you! In a revised version of the manuscript we will reference Figure 1 and the relevant subsection.

The description of the models was added to Sect. 2.2.2 and reference in Sect. 4.1.

L476: '...improving prediction skill' -> I assume you mean compared to your LSTM-baseline, right? Be more specific as the baseline comparison is important.

Indeed! We will add "in regard to the LSTM baseline" in a revised version of the manuscript.

The new statement now reads "... none of these constraints *improve prediction skill over the LSTM baseline*".

L485: 'These five basins were carefully chosen…' -> how? Please elaborate. Also, are these 5 basins the one in Fig. 6? This is not immediately clear. And follow up question on L519-520: Could this also be related to the hydrological processes involved in these catchments? You do not provide any information regarding the catchments, despite being carefully chosen (L485). Not all models can fit all catchments, and in the manuscript it is not explained how these catchments were selected/what are their characteristics and so on… This creates an uncertainty to the readers: how do we know that these results are not affected by the catchment selection and specific catchment behavior? In other words, can the authors ensure that their findings are transferable across and beyond the 5 selected catchments? (similar question - point for selecting the basin 73014 in L568 and Fig. 9).

We agree that this is not clear and will improve it in a revised version of the manuscript. The five examples were carefully chosen basins in which all hybrid models have performance on par with the LSTM, as in our didactic example. Moreover in Fig. 8 and Fig. 9 we use basin 73014 because it belongs to this subset.

Our study focuses on demonstrating what our proposed entropy metric can reveal rather than selecting optimal models; hence, it is not relevant what exactly are the characteristics in those catchments, but rather we see them as useful examples to demonstrate what the results of our proposed analysis may look like.

The choice of basins is now more clearly addressed in Sect. 4.2.

L564-L567: You mention the two different components. The first being, the low vs high entropy, and the second being the unaffected vs suspicious time-varying patterns. Do you think there could be a visual representation of this on a 2-axes plot? I am thinking something like figure 4 but 2-d. In this case, the more you move towards the 'high' entropy values, the less important the change in parameters is (you already have mentioned that high entropy indicates 'struggling due to the imposed constraint'), but the more you move towards the 'low' entropy values, you can get different 'color' (if we draw a comparison to fig. 4), depending on whether we have high or low variability in the parameter axis. This is not necessary, but I thought it might be interesting

to try and visualize this important insight - it could help promote and establish the detailed methodology.

Thank you for the interesting suggestion. The problem is that, as we saw in Section 3 and Figure 3, comparing the entropy in the parameters of models with different numbers of them will result in disingenuous conclusions. There are ways in which this could be avoided, but we find our methodology most simple by focusing just on the entropy of the data-driven component.

No changes were made to the manuscript based on this comment. The issue of comparing the entropy of a model's parameters and its hidden states was addressed as a reply to the reviewer.

L573-574: Wouldn't the fact that the model has learned behaviors from training on other basins a good thing? I am confused about the point you aim to make here.

This is clarified in the next sentence. For this basin, SHM by itself already proved to be a good model and there is really no benefit from the hybrid approach and training in multiple basins.

No changes were made to the manuscript based on this comment.

L585-589 and Figure 10: You mention this later on, but it would be nice to elaborate already a bit on why the Hybrid Nonsense is the top 2 - it is a logical question that a lot of readers would immediately have.

Agreed! In a revised version of the manuscript we will start to comment here.

Section 4.3.3 was modified to address this and other comments.

Fig 11: Would be helpful to add the % in the bars as well.

The relevant percentages are already used in the text and the bars already give a sense of proportion. Moreover, we used the same scale for the x-axis in Figure 11 and Figure 14, so we would argue that there is no need to include percentages in the figure.

As the percentages are already in the text, Fig. 11 was not modified.

General question about section 4: What about a joint analysis of ΔH and ΔNSE? I mentioned this earlier as well, but when trying to simulate real world catchments, the performance of the models can vary quite a lot - then it would be hard to judge and compare the different entropies across models, if the baseline of their predictive performance is not comparable. It would be nice to provide some clarifying perspectives on this while concluding section 4.

Thank you for this important point and, as we mentioned in our previous reply, a decision about model selection should account for performance and not only our proposed metric, which is tailored to diagnostics, not model selection in specific. In fact, with this manuscript we don't

make any specific statements about model selection because we believe that this is an even broader topic in which not only performance, and now entropy, should be considered, but also interpretability and computational cost, for example. We will, however, comment on possible ways to include ΔH and ΔNSE in a joint metric for model selection in a revised version of the manuscript, as "nuclei" for future research.

The introduction of Sect. 4.3.3 comments on using entropy as part of a set of criteria for model selection while not being a criterion on its own.

L648: I would like to suggest adding the word 'evaluating': 'building **and evaluating** hybrid models'. It is a bit 'nit-picky' as a comment, but I think it is an important part of your implications and deserves to be mentioned.

Thank you for this suggestion. We also see how the word 'evaluating' fits here and will add it in a revised version of the manuscript.

The statement now reads "These results overwhelmingly suggest that we need to reconsider our ways of building *and evaluating* …". Additionally, this statement appeared previously in Sect. 4.3.3 and removed from there.

L672: This sentence reads like you imply something along the lines of: 'process-based modeling of catchment scale streamflow is unnecessary - why go in the long effort of creating or applying these models if LSTMs can be better?'. I know this is not your initial intent, so I would suggest rephrasing in order to avoid confusion.

Indeed, that is not our intention. We will place this better into context in a revised version of the manuscript.

Section 4.3.3 and Sect. 5 were modified to consider this comment.

**Technical Corrections**

L38: 'Typically catchment scale processes of in a rainfall-runoff…'

L174: '...used [in] our case…

L210: us -> is?

L211: you repeat 'of'

L253: I believe 'setup' is just a noun - 'set [space] up' should be the verb needed in this context. Maybe I am wrong.

L351: 'fix' -> 'adjust' might be better fitting?

L408: is there a full stop [.] missing after 'model'?

L472-475: This period is 4 lines long and quite hard to read through - I would suggest splitting up to individual sentences to ensure readability.

General 'correction': make sure you adopt a unified style when it comes to capitalizing titles throughout the manuscript. I see some passages have a 'capitalized' style (e.g., 'Comparing Conceptual Constraints on the Entropy Axis') and some others are more free with capitalizing words (e.g., '3.3.2 Measuring entropy of conceptual model parameter space').

Thank you for your careful reading of our manuscript! In a revised version we will fix all of these technical corrections.

All of these changes were considered in the revised version of the manuscript.

**Review by Anonymous Reviewer #2**

1. The LSTM in the hybrid model is used for parameter estimation, while the simulation is conducted by the process-based model (PBM). The main reason for incorporating PBMs is to improve model interpretability and performance, especially when data is limited (by reducing the searching space). The latter is achieved through physical constraints provided by PBMs. The physical constraints include not only mass balance for each bucket but also the interactions between fluxes. More specifically, the hybrid model provides clearer expressions (better interpretability) of the physical processes than black-box models (e.g., LSTM) and enables the calculation of internal variables. This means that we can incorporate more data as target values to further constrain or evaluate the model.

Many studies, including those by the authors, showed that the internal variables such as soil moisture and ET are comparable to observations. If the physics are totally incorrect (or, as the authors commented --- "physics-ignored"), how is it possible that these internal variables make sense? If some of the modules carry realism to some extent (which would make "physics-ignored" incorrect), can the authors' method ascertain what is good or correct? If the method cannot ascertain what is good, how can it separate the good from the bad?

Currently, the model in this study is trained only with streamflow data and evaluated against streamflow. However, the authors could design a more complex synthetic case with additional internal variables such as ET, snow, soil moisture, and baseflow, and explore the following:

   a. If we add more constraints to the internal variables (by using them as additional targets), can the nonsense formulation still achieve the same performance as the correct model?
   b. Can a model with a nonsense formulation provide accurate simulations of these internal variables when trained only on streamflow?
      In real cases, even when we lack direct observations for some variables, we can still compare model outputs to satellite or reanalysis data to gain insights into the accuracy of the internal variables and model structure.
   c. When your hybrid models produce accurate predictions for both internal variables and streamflow, do you still observe an increase in entropy when adding physical models? Or, if entropy increases, can you still maintain the same level of accuracy?

Thank you for this comment. We would like to clarify that we use "ignored" rather than "incorrect" physics because this better describes what may happen in hybrid models. Physical laws such as conservation of mass and physical principles like the interaction of fluxes are indeed encoded in the model, but in some cases they might not be satisfied in the ways in which the modeller might expect.

If the prescribed constraints are not respected in the hybrid model, then also interpretability is not given as such. This emphasizes the need for and usefulness of our proposed method: we need to better understand if the hybrid model under investigation truly respects the physics as

intended (in that case, interpretation and consistency tests make sense and are meaningful), or if the data-driven component effectively changes the structure - up to essentially manipulating all processes, leaving only mass balance as effective physics constraint (in that case, interpretation based on the initially prescribed physical processes would be misleading and not suitable for advancing science).

We suspect that hybrid models, more often than not, show an effective structure that is notably different from the initially prescribed physics constraint. We hope to motivate the readership and community to perform extensive tests with our proposed methodology to confirm or dismiss this hypothesis.

The reason why so far this issue has not been widely documented might lie in the fact that even hybrid models that are constrained by "inadequate physics" have the capacity to show high correlations with unobserved variables. Our Hybrid Nonsense model demonstrates this clearly as was shown in Acuña Espinoza et al. (2024). Despite deliberately encoding nonsensical interactions in between its fluxes and storages, this model still achieved a correlation with soil moisture (from ERA5-Land data) of 0.85 that is nearly identical to the more physically reasonable Hybrid SHM with a correlation of 0.86. This shows that compensation by a neural network can make even poorly encoded knowledge of physics appear to work well. One possible explanation is that soil moisture dynamics are mostly driven by boundary conditions (rainfall - soil moisture increases, no rainfall - soil moisture decreases), making it not a particularly challenging test. It remains an open question how consistency checks should be designed to be truly informative in a physics context; our proposed method will help establish the ground for interpretation of such consistency checks by warning the modeler if the prescribed constraints are actively compensated for.

The key contribution of our work is hence providing a systematic way to distinguish when physics are consistently utilized versus ignored and manipulated. Our entropy metric, benchmarked against a purely data-driven baseline, enables this separation. Higher entropy than the baseline indicates "bad" physics, i.e. constraints that need compensation, while lower entropy indicates "good" physics, i.e. contributing meaningfully to model performance. "Good" and "bad" is used as abbreviation for these longer definitions in the manuscript and properly explained as such; if that is perceived as too bold, however, we would consider replacing those terms in the caption of Fig. 4.

Thank you for these suggestions. These are indeed questions we have considered ourselves and would be valuable to explore in future work. However, we believe it's important to maintain a focused scope for this study to ensure we provide a thorough answer to our central research question. The performance of hybrid models as such, and the best training strategy (single target vs. multi-objective) to increase it, are not of core interest here (as we noted above, the entropy metric is not to be confused with a performance score).

Furthermore, we do not wish this study and publication to become a comprehensive evaluation of the Nonsense model. The motivation for this study arose from the observation in Acuña

Espinoza et al. (2024) where good predictions of streamflow were achieved without the need to add a proper physics-based model like SHM, but using Nonsense instead. This led us to investigate why you could seemingly add any conceptual model structure, even deliberately "nonsensical" ones, and still achieve equally good hybrid predictions of streamflow. We believe our current scope allows us to provide a solid understanding of this phenomenon through both our synthetic case study and the CAMELS-GB case study, and we see the questions you've raised as natural and important extensions for future research.

Regarding the broader question of using correlations with internal variables to validate hybrid models: as discussed above, there is a need to increase the rigor of such analyses by putting the correlation results into the proper context. While important hybrid modeling studies have used correlation of internal variables as validation (Feng et al., 2022), from the other perspective, it has been shown that LSTM cell states can also achieve strong correlations with observed variables, even with individual cells presumably learning to track specific processes like snow accumulation (Kratzert et al., 2019).

Instead, we first need to understand what the hybrid model is doing, before we can interpret correlations. It might be that the internal functioning of the hybrid model is substantially different from what the modeler had put in, and hence a surprisingly high correlation with variables not used for training might just mean that the hybrid model found its way into an effective time-lagged representation of how rainfall increases the level of a storage, before producing discharge: this is what we observed in our test case with the Nonsense model. We therefore see our proposed analysis as a first and necessary step to put any correlation analysis on solid grounds. The result of a correlation analysis, however, will not have an impact on our method in any sense, which is why we don't see merit in extending the scope of the current manuscript to internal variables. Also, the question how we can better enforce physics constraints to prevent them from being overwritten is beyond the scope of this study; rather, the proposed method provides a tool for modelers to test the effectiveness of their way of constraining hybrid models.

In our reply to the reviewer's comment we described the issue of deriving interpretability from the imposed conceptual model constraint when this structure likely has been modified in the final hybrid model. We also described the issue of using correlation from unobserved variables with independent data products as validating. These points have been incorporated in multiple subsections of Sect. 1 and Sect. 5.

2. Should we make all parameters dynamic? Similar to traditional conceptual models, hybrid models also suffer from overfitting when too many parameters are calibrated (or "learned," in the case of hybrid models), especially when done dynamically. We should not make all parameters dynamic. This caution has been raised in several recent works on hybrid modeling. By making all parameters dynamic, we give the LSTM too much freedom and intentionally allow it to dominate the hybrid model. One key point I want to emphasize is that LSTMs (and neural networks in general) are "lazy" models—they tend to find the most convenient (easiest) way to make predictions. Rather than learning the true functional relationships of parameters, they

often memorize patterns in the time series, and often special effort is needed to make it extract the necessary information. To discourage this "laziness" in neural networks, two approaches can be considered:

    a. In addition to the temporal test as done in this work, prediction in ungauged basins or regions is a useful evaluation method. Verifying the accuracy of internal variables is another effective strategy. If a hybrid model demonstrates good spatial generalizability, how does its entropy compare to that of a pure LSTM?

    b. If we only allow a limited number of parameters to be dynamic, for example, just beta (shape coefficient in the soil moisture zone), can all models still achieve the same level of performance? How does their entropy compare to that of the pure LSTM?

The authors touch on a similar point in Lines 645–648: "Notably, this process occurs entirely under the hood. If we had evaluated performance using only NSE, we might have mistakenly concluded that the Nonsense constraint was just as valid as SHM or Bucket, since all three achieved the same performance when paired with the LSTM." I hope the authors can consider my suggestions in their entropy analysis experiments. The current experiments and narrative might unintentionally lead readers to conclude that the hybrid model lacks of effectiveness and interpretability.

Thank you for these suggestions. In fact, ultimately we wish to help improve the effectiveness and interpretability of hybrid models, by providing a rigorous analysis tool that helps looking under the hood. We challenge the assumption that hybrid models become effective simply because we add a specific structure like SHM. As reflected in the lines you highlighted, our intention is to demonstrate that, when predicting streamflow alone, some models might be good for the wrong reasons, and we aim to provide tools to identify such cases, to have a chance to correct them and develop better ways of building hybrid models.

We agree that not all parameters need to be dynamic, and our implementation actually provides the flexibility to keep certain parameters static while making others dynamic. Your suggestions represent excellent directions for future research. For example, if we freeze certain parameters, does the LSTM increase its reliance on remaining parameters (increasing entropy), or does it find more elegant solutions with the remaining degrees of freedom (reducing entropy)? These questions exemplify how this study opens new research avenues while advancing model evaluation beyond simple predictive performance to include model complexity, a topic that is typically not addressed.

Nevertheless, for this study we deliberately followed that of Acuña Espinoza et al. (2024), as we specifically wanted to address the questions those authors raised. To emphasize, we didn't do any new modeling for this study but analyzed previously published results.

With respect to prediction in ungauged basins: we focused on temporal rather than spatial extrapolation because we also followed the scope of Lees et al. (2021), which matches the standard benchmarking practice for CAMELS datasets. Theory-wise, we do not see the direct benefit of applying our entropy metric to the predictions of hybrid models in ungauged basins.

The potential overwriting of physics constraints happens during the training phase, and hence it is natural and logical to analyze it in the respective basins that these data are available for. In a secondary step, one could of course also use our method to analyze the variability of LSTM hidden states when predicting in ungauged basins, but the interpretation becomes more difficult - we see this as an open question for future work.

Regarding your point about neural networks being "lazy" models, please indulge us as we'd like to offer a philosophical perspective. We believe this apparent "laziness" sometimes aligns beautifully with the principle of Occam's razor in that what appears to be lazy may actually represent profound insight that recognizes that a complex problem can have a surprisingly simple solution. In our case, the LSTM's ability to ignore unnecessary complexity in models like SHM and focus on what's truly essential for streamflow prediction could be seen as elegant, efficient and parsimonious problem-solving rather than laziness. The fact that the LSTM could transform even our Nonsense model into an effective predictor suggests it identified the core requirements for this prediction task, potentially revealing important insights about what's actually necessary for predicting streamflow accurately in this dataset. Further, we designed the didactic synthetic examples deliberately such that they could reveal if LSTMs tend to do "overcomplicated" things or fail by only memorizing time patterns. It was reassuring to see that they identified the simplest and truest representation possible.

The suggestions made by the reviewer in this comment are appreciated and definitely provide avenues for future research. Our scope for this study is constrained by the scope of Acuña Espinoza et al. (2024) and we believe that we effectively have answered the question of why a model with a deliberately "nonsense" component proved effective at making predictions while introducing a metric the adds to a repertoire of criteria that can be used to assess hybrid models.

Based on this comment we made changes to several subsections of Sect. 4, especially Sect. 4.3.3, and Sect. 5 to acknowledge the limitations of this study and to better frame and give context to our results.

3. I also felt the entropy argument has some logical issues: The fact that LSTM has the lowest entropy simply means it has no reason to care about processes. Let's think of process-based Earth System models with land surface processes like energy balance and carbon cycles, with many complicated calculations and quite likely lower NSE values. Now, let's say we replace a component with large variability but minor influence on streamflow with an NN-based dynamic parameter. The dynamic parameter will learn some of the missing dynamics but will generate a large entropy with a quite small gain on streamflow performance, but does this mean it is worse than a model that completely ignores these dynamics? Moving one step further, if one "downstream component" to the NN has large structural error and generates lots of noise, the NN ends up ignoring this component (giving it nearly 0 weight), and route information through other paths. The total entropy may end up being lower. Does this mean the second one is a more realistic model?

Thank you for raising these questions.

To reiterate, our measure of entropy is designed to evaluate whether added model components contribute meaningfully to make the task of prediction easier or whether the neural network must compensate for their inadequacy. Entropy is used as a diagnostic tool. When we argue that entropy relates to the challenge of prediction (lines 394-401), we're focusing on this relationship between model complexity and difficulty in prediction. Further, as stated above, we do not wish to provide a model selection method that pits performance against entropy. Rather, we provide a tool that transparently shows how much the LSTM has to compensate, which is expected to be very valuable in model evaluation. We do not see the point of providing a universal recipe of how to balance this with performance. The main driver of advances is, from our perspective, understanding why our models perform as they do.

So coming back to your specific example, we believe that if a hybrid model learns to bypass structurally flawed components, this provides valuable diagnostic information. While this doesn't make the model more "realistic" in terms of process representation, it reveals which processes actually contribute to predictive skill versus which introduce unhelpful complexity. Understanding when and why processes are ignored can guide us toward better representations or help recognize when we're asking models to represent processes they cannot adequately constrain with available data.

No changes were made to the manuscript based on this comment. The issue of replacing processes or skipping a specific process were addressed in our reply to the reviewer.

4. One should not think that the NNs in a hybrid model learn only true physics --- it is a mixture of true missing (to be learned) processes and compensation for structural, parametric and even numerical deficiencies. The baked-in assumptions serve as constraints to limit the searchable subspace, hopefully making the framework more generalizable and giving meaning to intermediate variables. The hope is that the signal overwhelms the noise or that we can extract useful insights from learning. We verify the meaning using additional observations.

My point is that one should base such analysis on a case-by-case basis and the conclusion obtained by the experimental design in this study may not generalize to another case. Entropy measures variability, not correctness. A model may need high variability in parameters due to data noise, poor observability, or inherent system variability—not necessarily because the constraints are wrong. The authors acknowledge uncertainty later (Sect. 4.3.2 and 4.5) but still lean heavily on this deterministic interpretation throughout. Equating low entropy with model adequacy and high entropy with "physics being ignored" oversimplifies complex interactions between model structure, data, and learning dynamics.

Thank you for this comment. In rainfall-runoff modeling, where we work with significant abstractions of real-world dynamics, we acknowledge that what we learn are not "true physics" but rather macro-scale insights about system behavior, no matter if we talk about conceptual or hybrid models. Further, here the NN is not used to learn "true" physics, as these are assumed

to be encoded by the conceptual model which serves as the head layer of the LSTM. Rather, the LSTM learns how to bias-correct the conceptual constraint to arrive at close-to-true physics.

As the Reviewer states, models will struggle with data noise, poor observability, or unresolved system variability, and this will have an impact on how much effort the LSTM has to put in to make the rigid conceptual constraint perform well under these conditions. Hence, while it will be difficult to interpret individual entropy values as "high" or "low", the comparison with the reference entropy value of the pure LSTM, and potentially other hybrid model structures as done in our study, for a given catchment is highly informative. This is why we promote constructing at least parts of the model evaluation axis in Fig. 4, and why we look at basin-specific rankings in Fig. 11 to complement the results from simply comparing entropy values across basins in Fig. 10. It will be interesting indeed to further refine our understanding how "adversities" in the data of the basin to be modeled impact the difficulty of the modeling task and hence the entropy metric in future work.

We agree that evaluation should be case-by-case, which is why we promote a widely applicable method, not its analysis outcomes. We made a careful effort to both provide an overview of results for that specific large-sample data set, and additionally investigated specific basins in-depth for our analysis in Sections 4.2, 4.3.1, and 4.3.2. The purpose of these analyses is to let the readers gain intuition about the possible outcomes of our analysis method and their interpretation, such that they can apply the method to their cases of interest. It will be valuable to find out if our results generalize across many different large-sample datasets, and how they depend on the specific hybrid model architecture. Again, with this study, we hope to stimulate further research along these lines, to advance hybrid modeling in hydrology but as Reviewer 1 pointed out, as well in many other disciplines.

Similarly to comment no. 2, we made changes to several subsections of Sect. 4, and Sect. 5 to acknowledge the limitations of this study and to better frame and give context to our results.

5. Ignoring causality. None of the analyzed information metrics measure causality, while causality is a key value brought in with the incorporation of physics. The physical component can ensure that higher temperature can lead to higher evaporation, or ensuring that precipitation (not humidity) drives runoff. The causality and baked-in sensitivity allow future climate projections to be more reliable. Your information metrics do not reflect such guarantees.

Thank you for this observation. You are correct that our entropy-based analysis does not measure causality, because it is not designed to do so, and we also do not claim anywhere in the text that it would. We do agree that what the Reviewer describes as causality is a desirable property and insightful methods should be developed to measure/detect it in hybrid models. This is an open question, yet unrelated to the topic and goal of our study.

No changes were made to the manuscript based on this comment. Measuring causality is outside of the scope of this study.

Line 102–104: There are several other studies that have compared hybrid models with LSTM. These could also be mentioned here for completeness.

Thank you for this suggestion. Indeed there are, and we will include some additional references to comparisons between hybrid models and LSTMs in Section 1.3 of a revised version of the manuscript, while keeping these lines focused on motivating the current study.

Section 1.3 was expanded to include additional studies which compare hybrid models.

Line 451–453: This depends on which aspect of the problem you're examining. Hybrid models should be compared not only to LSTM but also to traditional process-based models. Given the same structure of a process-based model, the hybrid version can yield better performance. Compared to LSTM, hybrid models offer better interpretability. The fact that a hybrid model can make a nonsensical model perform well does not mean that a hybrid model built with a correct process-based structure is invalid. As I mentioned in my major comments, the model should be evaluated more comprehensively. Previous experience and domain expertise can help reduce equifinality problems and prevent obviously nonsensical model configurations.

Thank you for this comment. We agree that hybrid models may provide valuable capabilities between traditional and data-driven approaches, and we are not saying that well-designed hybrid models are invalid. As we have pointed out before, it is critical to understand if the prescribed physics are obeyed or not, before making a claim about interpretability.. The fact that our "nonsensical" structure could be transformed into an effective predictor suggests we should pay more attention to what is effectively happening in the hybrid model structure, and be more critical about whether what we're adding actually contributes to the task at hand. Our entropy analysis is intended as a complementary tool to help identify when neural networks are compensating for structural choices, allowing more informed decisions about the merits (such as interpretability) of a model.

Similar to the general comment, changes were made in Sect. 1.2, 1.3 and Sect. 5 based on this comment.

Line 561–562: This supports my view that LSTM is a "lazy" model. It tends to find ways to bypass physical constraints, especially when dynamic parameters are allowed.

While we wouldn't necessarily agree with the notion of "lazy" (we rather find it smart and reassuring that the LSTM finds a parsimonious, efficient and skillful representation of the system), we fully agree that the LSTM finds ways to bypass physical constraints if they are not helpful for the prediction task, and this is exactly what we shed light on with our method - in a quantitative way by means of our entropy metric, and qualitatively by analyzing the overwriting behavior of the dynamic parameters.

No changes were made to the manuscript based on this comment. The interpretation of LSTMs being "lazy" was addressed in our reply to the reviewer.

Line 637–638: A simplified task means that LSTM has found an easier path to make predictions, but this does not necessarily indicate higher accuracy. Multiple evaluation metrics should be used to support conclusions

We disagree with this point. All models were trained and evaluated using the same loss function (NSE or MSE), ensuring accuracy is directly comparable. So the LSTM indeed achieved higher accuracy in streamflow prediction. What we refer to as a "simplified task" is prediction with lower entropy: if the physics constraint took over main parts of prediction, the hidden states in the LSTM wouldn't have to vary as much as in the pure LSTM, and hence entropy would be lower. While additional metrics could provide more detail and further insight, our study focuses specifically on the "effort" of the LSTM in this hybrid model architecture.

No changes were made to the manuscript based on this comment.

---

## Referee Report (RR1)

I appreciate the authors' efforts in revising the manuscript and responding to the previous round of review. However, I think there are still some critical logical issues that remain unresolved:

**First**, in the synthetic case (Fig. 2), the central inference seems to be: conceptual model same as the "true model" or over-parameterized ⇒ low entropy; model different or under-parameterized ⇒ high entropy. But this does not mean that models with high entropy are "wrong," nor that low entropy means "correct." Entropy is not a sufficient condition for the adequacy of a model structure. Over-parameterized models can have lower parameter entropies. Hydrologic models with more process representations may require a larger subset of parameters to be dynamic to accommodate their complex structure for good performance, whereas simpler models may need only a few dynamic parameters. Does this imply that the latter has a better physical representation?

In addition, if only streamflow data from the "true model" is used to constrain the training (pretending this is the only information we know), we might think that Model 3 provides a better representation of hydrological processes since it has two-layer buckets (soil moisture and baseflow buckets) and gives an almost perfect streamflow prediction. However, when more information about the model structure or other variables is available, we see that this model is actually most different from the "true model." This simple example demonstrates why additional constraints are needed to diagnose whether a model reflects "bad" physics. More rigorous experiments, e.g. adding additional constraints to the training (as suggested in the previous round of review), are required to demonstrate how entropy can be meaningfully connected with the physical representation of a conceptual model. The authors argue that "we're focusing on this relationship between model complexity and difficulty in prediction." Can I then understand that entropy is related to model complexity, not to the correctness of the physical representation? However, what we need is not an easier model but a more physically correct model.

**Second**, the entropy of LSTM states might not reflect parameter entropy. As noted in the previous review, we should not make all parameters dynamic. For the same model, the selection of dynamic parameters will change the entropy. In addition, the parameter ranges differ between models in the synthetic case, which might make comparisons and diagnosis problematic. A hybrid model with flexible parameter ranges can still simulate well even with the wrong structure, since the governing equations of buckets the simple conceptual rainfall–runoff models are similar to each other (summarized to Eq. 1).

**Third**, the hybrid model using all static parameters could have similar performance to the one with some dynamic parameters in predictions for ungauged basins. I am not sure why the authors claim: "The potential overwriting of physics constraints happens during the

training phase, and hence it is natural and logical to analyze it in the respective basins that these data are available for." One can still calculate the entropy of the LSTM and parameters. In this case, can we say the model using all static parameters has a better physical representation than the one with dynamic parameters? However, they use the same physical model. How the NN part of the hybrid model is designed also matters. Again, equating low entropy with "adequate physics" and high entropy with "physics being ignored" oversimplifies the problem.

I think these issues, along with those raised in the previous round of review, need to be addressed in the current work; otherwise, the results could mislead readers. Unfortunately, the authors defer them to future work, which prevents me from supporting the publication of this manuscript in its current form.

---

## Author Response (AR2)

**Author's response**

Dear Dr. Fenicia,

Thank you for handling our manuscript and enabling our discussion with the reviewers.

To your specific points:

Reviewer 2 acknowledges improvements but finds critical logical issues remain, warranting rejection in the current form. There is a concern that entropy is oversimplified: low entropy does not guarantee physical adequacy nor high entropy inadequacy. The synthetic case needs added constraints, dynamic vs. static parameter handling requires clarification, and claims regarding hybrid models and physics overwriting are considered misleading.

We thank the editor for their assessment. We have considered Reviewer #2's concerns and responded systematically to each point as detailed below.

The core disagreement between the reviewer and ourselves is on scope. Reviewer #2 requests additional experiments (prediction in ungauged basins, alternative parameterizations, multivariable constraints) that were not part of the original study we are analyzing. Our paper is explicitly a diagnostic analysis of results from "To bucket or not to bucket" (Acuña Espinoza et al., 2024), and while our work certainly opens up new avenues for future research, we do not see the benefit of extending the scope of this manuscript too widely - its value and beauty lie in the straightforward application of a single, new metric that allows for deep diagnostic insights with our proposed analysis routine. Handling too many deviations from this setting will reduce the clarity of the manuscript to an extent that hinders true understanding of our method instead of adding to it.

Based on the review comments, we have clarified our methodology to emphasize the idea that: identical LSTM architectures across all models enable an unbiased comparison using entropy, and we explicitly frame entropy as one diagnostic tool within a larger evaluation framework, not as a universal metric of physical adequacy. Moreover, their concerns about all parameters being dynamic have been addressed in a new section.

Reviewer 1 agrees most concerns were addressed but emphasizes the need to justify the exclusive focus on streamflow prediction, incorporate discussion of equifinality, and ensure entropy metrics are not biased by differing model architectures.

We thank the editor and Reviewer #1 for their constructive feedback. We have addressed all three remaining concerns in the revised manuscript as detailed in our responses below. We believe these revisions strengthen the manuscript's clarity and methodological rigor, and are confident that any doubts have been resolved.

Considering these responses, we respectfully submit that Reviewer #2's requested changes represent a substantial expansion of scope beyond what we believe is graspable in an already reasonably lengthy single manuscript. While we have addressed all points that are within the manuscript's stated objectives and added a new section outside of them, implementing the remaining suggested changes would effectively require conducting a different study, moving away from our own objectives. We are further concerned that these changes might lead to an iterative cycle where new concerns arise, as the reviewer appears to envision a different study than what was originally in the preprint. Given that Reviewer #1 has recommended acceptance with minor revisions, that we have addressed Reviewer #2's concerns where feasible, and that some criticisms of Reviewer #2 are in contradiction to our results, we respectfully request that the editor evaluate whether the remaining points constitute revisions or represent a fundamental disagreement about research direction.

Kind regards,
Manuel Álvarez Chaves on behalf of the co-authors

**Report #1 by Georgios Blougouras & Shijie Jiang (co-review team)**

We thank the authors for responding to our comments. Most of our concerns have already been well addressed in the revised version of the manuscript. However, we believe that a handful of our aforementioned comments require additional revisions in order to ensure the quality of the manuscript.

We thank the co-review team for their feedback and address the remaining concerns in a revised version of the manuscript.

(following the labels of our initial review file)
General comment 2:
We still believe that there needs to be a clear explanation regarding why streamflow is the sole predictive focus of the manuscript. Streamflow (or any single-variable) prediction is not the only dominant application of HMs in the last years, and 'identifying correlations' (as the authors put it) does not paint the full picture; for example, current studies are actively using hybrid models to infer unseen variables and calibrate under a multi-task-learning approach (in an attempt to ensure process fidelity). We definitely agree with the added point in 1.4 (and the conclusions): by-passing the physical constraints indeed makes us concerned about the validity of inference. Yet, there is still no clear-in text clarification regarding why only single-output hybrid models are evaluated, as the scope of hybrid models can extend beyond single-variable performance. We believe this clarification would fit specifically in the Introduction, where the authors introduce the motivation for their study and their modeling choices.

Thank you for stressing this point. We recognize that hybrid modeling can extend beyond the prediction of single variables and the review process has made it clear that we should be more explicit about our rationale. We focus on predicting streamflow alone because we are evaluating the value of adding prior knowledge about the rainfall-runoff process in the form of conceptual models to an LSTM network. While we agree that if the goal is a hydrologically consistent hybrid model of high fidelity, multi-task-learning is the way to go, but it is by no means standard practice yet. Again, as highlighted in all versions of the manuscript and reiterated in the first revision, we are not promoting a specific approach to hybrid modeling, but offer an analysis for existing types of hybrid modeling that raise doubts - and this is exactly the setting of using conceptual models to apparently constrain the LSTM, for single-output prediction only. As suggested, we have clarified this motivation in the introduction, l. 154:
*"Note that we focus on a typical single-task prediction (here: streamflow) to evaluate the value of adding prior process knowledge (here: rainfall-runoff) in the form of conceptual models to an LSTM network. Yet, we recognize the potential of hybrid models for multi-task learning, where models are evaluated on multiple objectives including multiple target variables, and anticipate that our proposed method can be readily extended to such evaluations in future work."*

General comment 3:
Even though the authors aim to reserve the in-depth exploration of equifinality – entropy for a separate study, it nevertheless remains an issue that needs to be (at least partially) discussed,

as it is a core concern in hybrid modelling and hydrological models in general. The internal author responses to our comments are already a great step towards this direction - these (or equivalent explanations) should be added to the Introduction or the Discussion parts of the manuscript.

Thank you for this comment and it is reassuring to hear that our internal discussion is what you had in mind. We have now included the discussion of equifinality in the manuscript, both in section 3 relating to the synthetic examples (l. 495):

*"Interestingly, equifinality did not pose an issue with synthetic data in our experiments, as all models achieved perfect predictive performance and the model was always identifiable under the right conditions. This matches the experience of Spieler et al. (2020). However, in real-world applications, equifinality is likely to be more pronounced due to measurement errors, incomplete observations of the system under study, and other sources of uncertainty. This issue is discussed further in Sect. 4.3.2."*

and in section 4 relating to the CAMELS-GB case study (l. 656):

*"To return to the point of equifinality made in Sect. 3.4, as we have seen in this section, different hybrid model configurations may achieve similar predictive performance while exhibiting varying levels of entropy in the LSTM hidden state space and modifications to their internal behavior. We argue that high variability in parameter combinations represents an undesirable condition in terms of model structure specification. High entropy aligns with this perspective and, in general, entropy can be used to distinguish between equifinal models."*

Specific comment on section 2.3 ('I am concerned that the entropy metric... ML setup chosen in this study'):
Similar to the above point, we believe that the internal reply to the reviewer made by the authors should be incorporated in-text as well. The authors should instruct the readers on how to ensure that when they apply these entropy metrics themselves, these metrics do not end up being biased, e.g., from comparing models with different architectures.

Based on the reviewer's feedback, we have included a brief description of the setup in Sect. 3.2. More specifically, we provide more details regarding the choice of the ten hidden state LSTM and the number of basins (l. 324):

"The LSTM architecture of the baseline model and the hybrid models consists of ten hidden states. For our entropy analysis of the hidden states to be meaningful and fair, it is important to compare models of the same architecture. The choice of the number of hidden states was defined by the minimum required so that both the baseline model and hybrid models achieved equal performance. To aid in this process, the models were trained on a subset of five randomly selected basins (76005, 83004, 46008, 50008, and 96001) from the CAMELS-GB dataset.

(...)

We base our entropy analysis on equal performance, to ensure fair statements about the role of the conceptual component in a hybrid model ."

**Report #2 by Anonymous Reviewer #2**

I appreciate the authors' efforts in revising the manuscript and responding to the previous round of review. However, I think there are still some critical logical issues that remain unresolved:

**First**, in the synthetic case (Fig. 2), the central inference seems to be: conceptual model same as the "true model" or over-parameterized ⇒ low entropy; model different or underparameterized ⇒ high entropy. But this does not mean that models with high entropy are "wrong," nor that low entropy means "correct." Entropy is not a sufficient condition for the adequacy of a model structure. Over-parameterized models can have lower parameter entropies. Hydrologic models with more process representations may require a larger subset of parameters to be dynamic to accommodate their complex structure for good performance, whereas simpler models may need only a few dynamic parameters. Does this imply that the latter has a better physical representation?

In the synthetic case, indeed we found that overparametrized models had lower entropies but not lower than the "true" model. This is because for the overparametrized models there are degrees of freedom that the LSTM can adjust but which are not necessary.

We fundamentally disagree that "hydrologic models with more process representations may require a larger subset of parameters to be dynamic to accommodate their complex structure for good performance". Dynamic parameters, in our view, represent processes that one does not fully understand and therefore is letting a neural network fill this gap in understanding. If one could write a hydrological model using equations that describe each and every process in their full detail, there would be no need to couple this model with a neural network and therefore no entropy. The empirical results from the synthetic examples support this claim.

In addition, if only streamflow data from the "true model" is used to constrain the training (pretending this is the only information we know), we might think that Model 3 provides a better representation of hydrological processes since it has two-layer buckets (soil moisture and baseflow buckets) and gives an almost perfect streamflow prediction. However, when more information about the model structure or other variables is available, we see that this model is actually most different from the "true model." This simple example demonstrates why additional constraints are needed to diagnose whether a model reflects "bad" physics.

We respectfully disagree with this interpretation. To emphasize a point that we have argued before, we are evaluating the added benefit of a specific conceptual model in helping the hybrid model predict streamflow alone. For this specific example, without additional constraints, entropy shows that Model 3 does not provide a realistic interpretation of the synthetic truth (which has only one reservoir). As shown in Figure 4, Model 3 sits furthest to the right on the entropy axis relative to the data-driven model, whereas the true model is located furthest to the left. Further, the overparameterized models that can be reduced to the true model fall between the true model and the data-driven baseline. This positioning alone successfully diagnoses the inadequacy of Model 3's physical representation in characterizing the processes required to generate streamflow. This is precisely the diagnostic capability we propose. Our approach is fully consistent in interpretation and does not require "additional constraints" to "diagnose whether a model reflects 'bad' physics".

More rigorous experiments, e.g. adding additional constraints to the training (as suggested in the previous round of review), are required to demonstrate how entropy can be meaningfully connected with the physical representation of a conceptual model. The authors argue that "we're focusing on this relationship between model complexity and difficulty in prediction." Can I then understand that entropy is related to model complexity, not to the correctness of the physical representation? However, what we need is not an easier model but a more physically correct model.

Both in the paper and in our previous answer we refer to the complexity of the LSTM measured by the entropy of its hidden states. If the conceptual model adequately represents the generation process of streamflow in the catchment, the LSTM will have low entropy/low complexity. If the conceptual model doesn't accurately represent the streamflow generation process then the LSTM has high entropy/high complexity. Therefore it is the correctness of how the conceptual model represents the physical process.

**Second**, the entropy of LSTM states might not reflect parameter entropy. As noted in the previous review, we should not make all parameters dynamic. For the same model, the selection of dynamic parameters will change the entropy. In addition, the parameter ranges differ between models in the synthetic case, which might make comparisons and diagnosis problematic. A hybrid model with flexible parameter ranges can still simulate well even with the wrong structure, since the governing equations of buckets the simple conceptual rainfall–runoff models are similar to each other (summarized to Eq. 1).

We appreciate the reviewer's considerations regarding measuring entropy and model comparison.

As shown with the synthetic examples, the LSTM states are an unbiased way to analyze the entropy of the parameters and to compare between models with different numbers of parameters and different value ranges. Each hybrid model is coupled with an LSTM of identical architecture. While the conceptual models differ, they share this common LSTM component, which provides a consistent basis for comparing entropy across models.

Regarding dynamic parameter selection, this paper analyzes results from "To bucket or not to bucket" (Acuña Espinoza et al., 2024), which established the hybrid modeling framework with all parameters made dynamic. The current work focuses on diagnosing and interpreting those results through entropy analysis, rather than conducting new experiments with alternative architectures or parameterizations. Further, we do not see the fundamental difference to the approach we have proposed: if only specific parameters are made dynamic, then it is among these parameters to compensate for any misspecification in the conceptual constraint, so the individual entropies per parameter are expected to increase, and it is joint entropy over all hidden states that we measure, so we still see entropy as a fully consistent and fair metric.

To exemplify this, we have added a new Section 4.4 to the manuscript, with the previous Section 4.4 moved to Section 4.5. Subplot a) of the new Figure 12 shows the NSE CDF curves for the LSTM, SHM, Hybrid SHM, and a new model variant: SHM with only the parameter β in the soil moisture reservoir made dynamic (7 static parameters, 1 dynamic parameter). As expected, this single-parameter dynamic model shows a slight performance increase compared to the fully static conceptual model, but does not match the performance of the model with all dynamic parameters.

Subplot b) presents the entropy distributions for these models. The fully static conceptual model has no entropy and therefore is not shown in the figure. The single dynamic parameter variant exhibits an increase in entropy, even higher than the Hybrid SHM model with all parameters dynamic. This observation is consistent with our framework: when the LSTM must compensate for model misspecification through only one degree of freedom instead of eight, its activity (and thus entropy) increases substantially without proportional performance gains. To put it in different words, for SHM β dynamic, the LSTM has to "fix" the entire conceptual model through a single point of entry. By contrast, in the Hybrid SHM case where the LSTM is able to make smaller adjustments across multiple model components, this results in higher performance and lower entropy.

This example demonstrates that entropy serves as a diagnostic for how the neural network component compensates for structural inadequacies in the conceptual model. One can imagine additional experiments where parameters are selectively made dynamic or static to diagnose individual components in the conceptual model, with entropy guiding this process toward a representation that most accurately matches the natural system.

The reviewer correctly notes that parameter ranges differ between models. However, our entropy metric is derived from LSTM hidden states, not directly from parameter values. As explained previously, this provides a normalized comparison framework where the consistent LSTM architecture ensures comparability despite differences in the underlying conceptual models.

**Third**, the hybrid model using all static parameters could have similar performance to the one with some dynamic parameters in predictions for ungauged basins. I am not sure why the authors claim: "The potential overwriting of physics constraints happens during the training phase, and hence it is natural and logical to analyze it in the respective basins that these data are available for." One can still calculate the entropy of the LSTM and parameters. In this case, can we say the model using all static parameters has a better physical representation than the one with dynamic parameters? However, they use the same physical model. How the NN part of the hybrid model is designed also matters. Again, equating low entropy with "adequate physics" and high entropy with "physics being ignored" oversimplifies the problem.

I think these issues, along with those raised in the previous round of review, need to be addressed in the current work; otherwise, the results could mislead readers. Unfortunately, the authors defer them to future work, which prevents me from supporting the publication of this manuscript in its current form.

We appreciate the reviewer's detailed concerns and address them systematically below.

Regarding predictions in ungauged basins (PUB), in our previous response, we focused on the fact that the overwriting is learned during training, so it makes sense to us to investigate this behavior for those basins with available data. However, we agree that one can *additionally* analyze the entropy when using the trained model to predict in an ungauged setting, to assess the amount of overwriting happening - it just doesn't have the 1:1 translation of overwriting that *had* to happen to achieve good performance in fitting data, which is the core interest of our study.

Nevertheless, this paper is a follow-up to "To bucket or not to bucket" (Acuña Espinoza et al., 2024), which did not include PUB experiments, and we do not see the merit of extending our analysis to this setting because its interpretation does not add anything more didactic than what we have shown so far.

Outside of the ungauged setting, we do see the question of comparing models with dynamic versus static parameters as interesting. In fact, we observe a few instances where conceptual models with static parameters achieve performance equivalent to their dynamic counterparts. For example,

Basin 85001:

| Model | NSE | Entropy |
|---|---|---|
| LSTM | 0.921 | -119.16 |
| Hybrid SHM | 0.923 | -133.26 |
| SHM | 0.900 | |

Basin 94001:

| Model | NSE | Entropy |
|---|---|---|
| LSTM | 0.935 | -121.26 |
| Hybrid SHM | 0.957 | -128.56 |
| SHM | 0.926 | |

In these cases, the modest performance improvement of the hybrid model over the static version, combined with entropy values lower than the data-driven baseline, suggests adequate physical representation. The hybrid framework with dynamic parameters provides only marginal benefit for these particular basins. We allude to these specific examples in Section 4.3.1 and, more specifically, in Figure 7.

Concerning the influence of neural network architecture on entropy, we reiterate that all models in our comparison use identical LSTM architectures. This controlled setup allows us to attribute differences in entropy to the conceptual model component, as explained in our response to the previous point.

As a final point, we respectfully disagree with the reviewer's last statement saying that our interpretation oversimplifies the problem. Our framework interprets entropy as follows: low entropy indicates that the conceptual model provides an adequate physics-based representation that effectively constrains the hybrid model's behavior. High entropy indicates that the conceptual model misrepresents aspects of the natural system, requiring the neural network component to compensate through adjustments in the conceptual model's parameters to achieve accurate predictions. In such cases, the success of the hybrid model should be attributed primarily to the neural network's flexibility rather than to the physics-based conceptual structure. This diagnostic capability: distinguishing whether good performance stems from appropriate physical structure or from a neural network that compensates, is what entropy reveals and what we aim to communicate.

**References**

- Acuña Espinoza, E., Loritz, R., Álvarez Chaves, M., Bäuerle, N., & Ehret, U. (2024). To bucket or not to bucket? Analyzing the performance and interpretability of hybrid hydrological models with dynamic parameterization. *Hydrology and Earth System Sciences*, *28*(12), 2705–2719. https://doi.org/10.5194/hess-28-2705-2024

- Spieler, D., Mai, J., Craig, J. R., Tolson, B. A., & Schütze, N. (2020). Automatic Model Structure Identification for Conceptual Hydrologic Models. *Water Resources Research*, *56*(9), e2019WR027009. https://doi.org/10.1029/2019WR027009